Manuscript prepared for Atmos. Chem. Phys.
with version 2014/09/16 7.15 Copernicus papers of the LATEX class copernicus.cls.
Date: 13 October 2017

# Ozone sensitivity to varying greenhouse gases and ozone-depleting substances in CCMI-1 simulations

Olaf Morgenstern[1], Kane A. Stone[2,a], Robyn Schofield[2], Hideharu Akiyoshi[3],
Yousuke Yamashita[3,b], Douglas E. Kinnison[4], Rolando R. Garcia[4], Kengo Sudo[5],
David A. Plummer[6], John Scinocca[7], Luke D. Oman[8], Michael E. Manyin[8,9],
Guang Zeng[1], Eugene Rozanov[10,11], Andrea Stenke[11], Laura E. Revell[11,12],
Giovanni Pitari[13], Eva Mancini[13,14], Glauco Di Genova[14], Daniele Visioni[13],
Sandip S. Dhomse[15], and Martyn P. Chipperfield[15]

[1]National Institute of Water and Atmospheric Research (NIWA), Wellington, New Zealand
[2]School of Earth Sciences, U. Melbourne, Victoria, Australia
[3]National Institute of Environmental Studies (NIES), Tsukuba, Japan
[4]National Center for Atmospheric Research (NCAR), Boulder, Colorado, USA
[5]Graduate School of Environmental Studies, Nagoya University, Nagoya, Japan
[6]Environment and Climate Change Canada, Montréal, Canada
[7]CCCMA, University of Victoria, Victoria, Canada
[8]NASA Goddard Space Flight Center, Greenbelt, Maryland, USA
[9]Science Systems and Applications, Inc., Lanham, Maryland, USA
[10]Physikalisch-Meteorologisches Observatorium Davos – World Radiation Center, Davos, Switzerland
[11]Institute for Atmospheric and Climate Science, ETH Zürich, Zürich, Switzerland
[12]Bodeker Scientific, Christchurch, New Zealand
[13]Dipartimento di Scienze Fisiche e Chimiche, Università dell'Aquila, L'Aquila, Italy
[14]CETEMPS, Università dell'Aquila, L'Aquila, Italy
[15]School of Earth and Environment, University of Leeds, Leeds, UK
[a]now at: Massachusetts Institute of Technology (MIT), Cambridge, Massachusetts, USA
[b]now at: Japan Agency for Marine-Earth Science and Technology (JAMSTEC), Yokohama, Japan

*Correspondence to:* Olaf Morgenstern (olaf.morgenstern@niwa.co.nz)

**Abstract.** Ozone fields simulated for the first phase of the Chemistry-Climate Model Initiative (CCMI-1) will be used as forcing data in the $6^{th}$ Coupled Model Intercomparison Project. Here we assess, using reference and sensitivity simulations produced for CCMI-1, the suitability of CCMI-1 model results for this process, investigating the degree of consistency amongst models regarding their responses to variations in individual forcings. We consider the influences of methane, nitrous oxide, a combination of chlorinated or brominated ozone-depleting substances, and a combination of carbon dioxide and other greenhouse gases. We find varying degrees of consistency in the models' responses in ozone to these individual forcings, including some considerable disagreement. In particular, the response of total-column ozone to these forcings is less consistent across the multi-model ensemble than profile comparisons. We analyze how stratospheric age-of-air, a commonly used diagnostic of stratospheric transport, responds to the forcings. For this diagnostic we find some salient differences in model behaviour which may explain some of the findings for ozone. The findings imply that the ozone fields derived from CCMI-1 are subject to considerable uncertainties regarding the impacts of these anthropogenic forcings. We offer some thoughts on how to best approach the problem of generating a consensus ozone database from a multi-model ensemble such as CCMI-1.

## 1 Introduction

The Chemistry-Climate Model Initiative (CCMI), in its first phase, has produced an unprecedented wealth of simulations by 20 chemistry-climate and chemistry-transport models (Eyring et al., 2013). All of them comprise interactive chemistry schemes focussed on the simulation of stratospheric and/or tropospheric ozone, but there are significant differences in their formulations that affect chemistry as well as many other aspects (Morgenstern et al., 2017). One purpose of CCMI-1 is to inform the upcoming $6^{th}$ Coupled Model Intercomparison Project (CMIP6; Eyring et al., 2016), and particularly to provide pre-calculated ozone climatologies to those CMIP6 General Circulation Models (GCMs) that do not simulate ozone interactively. This is complicated by significant inter-model differences amongst the CCMI-1 models as well as the fact that CMIP6 will explore a variety of Shared Socio-economic Pathways (SSPs; Riahi et al., 2016) that expand on the Representative Concentration Pathways (RCPs; Meinshausen et al., 2011) forming the basis of CMIP5 and CCMI-1. Hence there is a requirement for a robust mechanism to turn the CCMI-1 ozone fields into merged climatologies that are consistent with those SSPs. The feasibility of this processing step hinges upon the degree of consistency with which the CCMI-1 models respond to variations in forcing fields; this is the topic of the present paper. More generally, the presence of targeted sensitivity simulations in the CCMI-1 ensemble allows us to study in detail the model responses to forcings by individual gases, which are of significant scientific interest irrespectively of applications in CMIP6.

Here we only assess the model responses to long-lived gas forcings. Regarding short-lived climate agents, there are large inter-model differences in the representation of tropospheric ozone chemistry

(Morgenstern et al., 2017) as well as spatially very heterogeneous emissions of ozone precursors. Due to these additional complexities, comprehensively assessing the consistency of the simulation of tropospheric ozone in CCMI-1 models needs to be the topic of a separate paper. Notwithstanding this, large-scale global climate and composition change can influence surface ozone through in-situ chemistry, long-range transport, stratosphere-troposphere exchange, changes in temperature and humidity, and radiative transfer.

We consider separately the influences of the following four different anthropogenic forcings on ozone ($O_3$): methane ($CH_4$), nitrous oxide ($N_2O$), ozone-depleting substances (ODSs, comprising chlorofluorocarbons, other organic chlorine compounds, methyl bromide, halons, and other organic bromine compounds), grouped together as "equivalent chlorine" ($Cl^{eq}$), and a group of greenhouse gases (GHGs) comprising $CO_2$ and fluorinated compounds (hydrofluorocarbons, HFCs, perfluoro-carbons, PFCs, and sulfur hexafluoride, $SF_6$) that do not act as ODSs. These gases are grouped together here as "$CO_2$-equivalent" ($CO_2^e$) using the ratios of their "radiative efficiencies" to that of $CO_2$ (table 2.14 of IPCC, 2007) as conversion factors. All of these influences have been studied before (see below), but not all of them in a multi-model context. In all cases these forcings have both direct radiative (as GHGs) and chemical impacts. For the RCPs, the combined radiative impacts of GHGs can be summarized as warming the troposphere and cooling the stratosphere, with associated dynamical consequences, but the chemical impacts are more complicated and also induce secondary effects such as perturbations to stratospheric water vapour and ozone which themselves link to dy-namics. This complexity opens up the potential for differences in model behaviour, the topic of this paper.

Several previous studies have investigated the linkages between $CH_4$ and $O_3$ (e.g., Stevenson et al., 2000; Prather et al., 2001; Revell et al., 2012a; Morgenstern et al., 2013; Naik et al., 2013; Voulgarakis et al., 2013). Generally, these studies have found that methane increases lead to ozone increases in most of the lower and middle atmosphere (below 1 hPa) which amplify the global warming associated with methane. These increases are associated with a few different mechanisms, including methane's role as an ozone precursor in the troposphere and a slow-down of chlorine-catalyzed ozone depletion by $Cl + CH_4 \rightarrow HCl$. Since IPCC (2007), this link between $CH_4$ and $O_3$ has been accounted for by stating an effective global warming potential for $CH_4$ that takes into account those chemical feedbacks, also due to stratospheric water vapour production by methane oxidation. We will assess here the consistency to which the methane-ozone link is simulated in CCMI-1 models.

The impact of $N_2O$ on $O_3$ is thought to be well understood (e.g., Portmann et al., 2012; Revell et al., 2012b; Stolarski et al., 2015). $N_2O$ is generally chemically inactive in the troposphere. In the stratosphere it decays to form nitrogen oxides ($NO_x = NO + NO_2$) in a minor loss channel. $NO_x$ then participates in catalytic ozone depletion (Brasseur et al., 1999). It is the third most important

anthropogenic greenhouse gas after $CO_2$ and $CH_4$ (IPCC, 2007) and is now the leading ODS by emissions (Ravishankara et al., 2009).

The impact of organic halogens on stratospheric ozone is likewise well understood (for a review see Solomon, 1999). Essentially, these gases rise into the stratosphere where they release their halogen atoms which then engage in ozone depletion. This is particularly pronounced in the polar regions where chlorine is "activated" on polar stratospheric clouds, causing the Antarctic ozone hole to form (Farman et al., 1985) and also causing usually less severe but highly variable ozone depletion in the Arctic. This means their chemical impacts occur mostly in the "chlorine layer" around 40 km and in the lower stratosphere over the poles (Brasseur et al., 1999). However, through dynamical feedbacks, transport, and impacts on ultraviolet and longwave radiation such ozone depletion affects atmospheric composition throughout the troposphere and stratosphere (Madronich and Granier, 1992; Madronich, 1993; Fuglestvedt et al., 1994, 1995; Morgenstern et al., 2013). Southern-Hemisphere climate change is thought to have been dominated in recent decades by ozone depletion (for a review see Thompson et al., 2011), but there is limited evidence for an effect of Arctic ozone depletion on the Northern-Hemisphere circulation (Morgenstern et al., 2010). Under the Montreal Protocol, halogen-catalyzed ozone depletion is anticipated to reverse (WMO, 2014); a recovery of the Antarctic ozone hole is now unambiguously identified in observations (Solomon et al., 2016).

For analysis purposes, the ODSs are combined into a single index, equivalent chlorine ($Cl^{eq}$), which is the sum of all chlorinated and brominated organic compounds as imposed at the Earth' surface, weighted by the number of halogen atoms per molecule and multiplied by 60 for brominated compounds (Newman et al., 2007). $Cl^{eq}$ excludes here di- and tribromomethane ($CH_2Br_2$, $CHBr_3$) which significantly impact stratospheric ozone levels (Oman et al., 2016). They are imposed as invariant constants (Morgenstern et al., 2017) and hence are thought not to contribute to any trends. $Cl^{eq}$ is shifted by 4 years relative to the A1 scenario (WMO, 2014) to represent the time it takes for the turn-around in halogens caused by the implementation of the Montreal Protocol to propagate to middle and high latitudes of the stratosphere.

Finally, the gases grouped as $CO_2^e$, comprising $CO_2$, hydrogenated fluorocarbons (HFCs), perfluorocarbons (PFCs), and $SF_6$, are not thought to have a significant direct chemical impact on ozone, but as greenhouse gases have substantial impacts on temperature, humidity, and circulation, which in turn affect ozone (IPCC, 2013). Under the REF-C2 scenario assumed here (which merges RCP 6.0 for non-ODSs with the WMO (2011) A1 scenario for ODSs), the fluorinated gases do not contribute much to global warming, i.e. the reference simulations described below assume moderate emissions of them (Meinshausen et al., 2011). $CO_2$, the leading gas in this group, undergoes roughly a doubling between 1960 and 2100 in this scenario. Morgenstern et al. (2017) show graphs of all the long-lived forcings used here. While these gases, for the purposes of this paper, are combined into one measure ($CO_2^e$), their actual treatment varies by model, with some models considering or not considering certain minor GHGs in their radiation schemes (Morgenstern et al., 2017). Some others

use lumping which in itself has certain limitations. For example, increases in $CO_2$ are cooling the stratosphere whereas increases in HFCs would warm it (Hurwitz et al., 2015), meaning that $CO_2$ is not a perfect analogue for HFCs in our model simulations. However, simulations that would target separately the impacts of HFCs do not exist in the CCMI-1 ensemble.

In this paper, we assess the degree of consistency found across the CCMI-1 ensemble w.r.t. the impact of these forcings on ozone. We will do so by using sensitivity simulations performed for CCMI-1. One limitation of this approach is that it does not account for nonlinear interactions between the forcings (e.g., stratospheric cooling caused by $CO_2$ slows down gas-phase ozone depletion Portmann et al., 2012; Dhomse et al., 2016). We will address this further in section 7.

## 2  Models and data

### 2.1  Experiments used in this paper

Here we use simulations performed under the following experiments as requested for CCMI-1. The simulations generally cover 1960-2100 unless stated otherwise (Eyring et al., 2013; Morgenstern et al., 2017):

- REF-C2: In this experiment, GHGs, $CH_4$, and $N_2O$ follow the RCP 6.0 scenario (Meinshausen et al., 2011), and ODSs follow the A1 scenario of WMO (2014).

- SEN-C2-fCH4: Same as REF-C2, except $CH_4$ is held fixed at its 1960 value (Hegglin et al., 2016).

- SEN-C2-fN2O: Same as REF-C2, except $N_2O$ is held fixed at its 1960 value (Hegglin et al., 2016).

- SEN-C2-fODS: Same as REF-C2, except all chlorinated and brominated ODSs are held at their 1960 values.

- SEN-C2-fGHG: Same as REF-C2, except $CO_2$, $CH_4$, $N_2O$, and other non-ozone depleting GHGs are held at their 1960 values.

- SEN-C2-RCP26/45/85: Same as REF-C2, except the GHGs, $CH_4$ and $N_2O$ follow the RCP 2.6, 4.5, or 8.5 scenarios (Meinshausen et al., 2011). These simulations cover 2000-2100.

SEN-C2-fCH4, SEN-C2-fN2O, SEN-C2-fODS, and SEN-C2-fGHG simulations address the sensitivities to individual forcings, whereas the SEN-C2-RCP experiments assess the impacts of the variant RCP scenarios that can be seen as simultaneous variations of multiple forcings relative to the reference simulation. For example, we use RCP 8.5 here because it is characterized by the largest anthropogenic forcings. In particular, $CH_4$ growth is much more pronounced than in REF-C2 / RCP 6.0 (Meinshausen et al., 2011).

## 2.2 Models used in the paper

We use CCMI-1 model simulations for which ozone has been archived for REF-C2 and any of the other 4 sensitivity experiments. For the assessment of the influences of GHGs, we require simulations covering REF-C2, SEN-C2-fGHG, SEN-C2-fCH4, and SEN-C2-fN2O (see below). Table 1 lists the models and the number of simulations used for the sensitivity analysis in section 3. ACCESS-CCM

Table 1: Models used in this paper, with associated ensemble sizes of CCMI-1 simulations conducted.

| Model | reference | REF-C2 | fCH4 | fN2O | fODS | fGHG | RCP26 | RCP45 | RCP85 |
|-------|-----------|--------|------|------|------|------|-------|-------|-------|
| ACCESS-CCM | Stone et al. (2016) | 2 | | | 2 | | | | |
| CCSRNIES-MIROC3.2 | Akiyoshi et al. (2016) | 2 | 1 | 1 | 1 | 1 | 1 | 1 | 1 |
| CESM1-WACCM | Garcia et al. (2017) | 3 | 1 | 1 | 3 | 3 | | 1 | 3 |
| CHASER-MIROC-ESM | Sekiya and Sudo (2014) | 1 | 1 | 1 | 1 | 1 | | | |
| CMAM | Scinocca et al. (2008) | 1 | 1 | 1 | 1 | 1 | 1 | 1 | 1 |
| GEOSCCM | Oman et al. (2013) | 1 | 1 | 1 | | | | | |
| NIWA-UKCA | Morgenstern et al. (2009) | 5 | 1 | 1 | 2 | 3 | | | |
| SOCOL3 | Stenke et al. (2013) | 1 | 1 | 1 | | | | | |
| ULAQ-CCM | Pitari et al. (2014) | 2 | 2 | 2 | 1 | 1 | 1 | 1 | 1 |
| UMSLIMCAT | Tian and Chipperfield (2005) | 1 | 1 | 1 | 1 | 1 | | 1 | 1 |

145

also conducted two SEN-C2-fGHG simulations, but because of the missing SEN-C2-fCH4 and and SEN-C2-fN2O simulations, these will not be considered here.

These ten models are described by Morgenstern et al. (2017) and references therein. Except for ACCESS-CCM and NIWA-UKCA, they all use hybrid-pressure (or actual pressure, in the case of ULAQ-CCM) as their vertical coordinate. ACCESS-CCM and NIWA-UKCA use hybrid-height levels. Apart from differences in coupling (ACCESS-CCM is an atmosphere-only model, whereas NIWA-UKCA includes a deep ocean), these two models are identical. In the following, where we display vertically resolved results from these two models, these will be based on fields interpolated onto a 126-level grid, equally spaced in $\log p$ and spanning 1000 to 0.01 hPa. The underlying pressure climatology is taken from a NIWA-UKCA REF-C2 simulation.

The CCSRNIES-MIROC3.2 simulations were conducted on two different computers (REF-C2 (1), SEN-C2-fODS, SEN-C2-fGHG, and SEN-C2-RCP85 on an NEC SX9 machine, and REF-C2 (2), SEN-C2-fCH4, and SEN-C2-fN2O on an NEC SX-ACE). This resulted in some differences between the two REF-C2 simulations. We have therefore repeated all calculations detailed below now assuming that the CCSRNIES-MIROC3.2 simulations represent two different models. The results are essentially unchanged versus what is presented here. Hence for the purposes of this paper, CCSRNIES-MIROC3.2 is treated as one model.

UMSLIMCAT and CCSRNIES-MIROC3.2 have prescribed or only partially interactive tropospheric composition (Morgenstern et al., 2017) This affects the sensitivity of total-column to the external forcings considered here.

There are numerous differences in the formulations of the models that influence how they respond to external forcings. Stratospheric gas-phase chemistry is handled relatively consistently by the models. For example, their chemistry schemes all include ozone depletion by the $HO_x$, $NO_x$, $ClO_x$, and $BrO_x$ loss cycles, with rates taken from compilations such as Sander et al. (2011). Differences exist in the treatment of heterogeneous chemistry on polar stratospheric clouds. Also photolysis is handled in various different ways by the models, and there are differences in dynamics that also impact on how these models respond to external forcings (Morgenstern et al., 2017). We will present a limited analysis of how stratospheric age-of-air, a salient diagnostic often used to characterize stratospheric transport, relates to the responses in ozone produced by the models. A comprehensive analysis of which aspects of the models' formulation is responsible for differences in behaviour is however beyond the scope of this paper.

### 2.2.1 Method of analysis

We form zonally averaged ozone on model levels as represented by the CCMI-1 models. Next, we perform a linear expansion around the reference case defined by REF-C2. This means

$$\Delta O_3 = a\Delta CH_4 + b\Delta N_2O + c\Delta Cl^{eq} + d\Delta CO_2^e + \epsilon. \tag{1}$$

Here, $\Delta O_3$ is the difference in zonal-mean simulated ozone between two different scenarios, $\Delta CH_4$ and $\Delta N_2O$ are the differences in surface methane and nitrous oxide, respectively, and $\Delta CO_2^e$ and $\Delta Cl^{eq}$ are the differences in surface carbon dioxide-equivalent and equivalent chlorine as defined above.

$a$, $b$, $c$, and $d$ are determined using least-squares linear regression. Functions of latitude, level, and month of the year, they minimize the residual $\epsilon$. For example, to determine $a$ we use the difference in the zonal-mean ozone fields from REF-C2 and SEN-C2-fCH4:

$$\Delta O_3 = a\Delta CH_4 + \epsilon \tag{2}$$

and determine $a$ by regressing, at every latitude, model level, and month, the 140- or 141-year timeseries of $\Delta O_3$ against the same-length timeseries of $\Delta CH_4$, which is the global-mean surface methane mixing ratio as defined under RCP 6.0 minus its value in 1960. Equivalent analyses yield $b$, using REF-C2 and SEN-C2-fN2O, and $c$, using REF-C2 and SEN-C2-fODS. The SEN-C2-fGHG simulation keeps all GHGs including $CH_4$ and $N_2O$, but excluding ODSs, fixed at their 1960s levels. To account for the effects of fixing $CH_4$ and $N_2O$, we form a modified ozone field

$$O_3' = O_3(\text{SEN-C2-fGHG}) + a\Delta CH_4 + b\Delta N_2O \tag{3}$$

which is derived from the ozone field produced by the SEN-C2-fGHG experiment, $O_3$(SEN-C2-fGHG), but with the impacts of differences in $CH_4$ and $N_2O$ added. We then use the difference $\Delta O_3 = O_3(\text{REF-C2}) - O_3'$ in our regression analysis as before to determine $d$.

In this formulation, the forcings (except $Cl^{eq}$) are as imposed at the surface, so transport-related delays are not accounted for. Such delays primarily result from the time it takes for a long-lived tracer, emitted at the surface, to reach the stratosphere. For the forcings other than $Cl^{eq}$ this is not critical as their tendencies are only slowly varying, i.e. they do not display the sharp turn-around characterizing $Cl^{eq}$.

In cases where multiple simulations are available for a given scenario and model, the ensemble average is used in the analysis.

In the below, we only display the coefficients $a$, $b$, $c$, or $d$ where these are significantly (at the 95% confidence level) different from 0. Details on this process are in the appendix.

## 3 Results

### 3.1 Sensitivity of ozone to methane

Figure 1 shows the sensitivity of zonal-mean ozone with respect to changes in $CH_4$ (i.e., $a$) as derived from the REF-C2 and SEN-C2-fCH4 experiments. Nine models have conducted both experiments. The models agree on some general features of the signal, namely an increase of ozone in much of the lower and middle atmosphere, and a decrease in the mesosphere. In the middle and upper stratosphere, in all models there is a region where $CH_4$ increases cause ozone increases by around 10% to 40% of the increase of the prescribed surface methane mixing ratio. This may be because of the $CH_4 + Cl \rightarrow HCl$ reaction which returns chlorine to HCl not involved in ozone depletion. Higher up, above the stratopause at approximately 1 hPa, methane increases cause ozone to decline, due to increases in $HO_x$ related ozone depletion under increasing methane (Morgenstern et al., 2013, and references therein). There is considerable uncertainty regarding the size of this feedback. CCSRNIES-MIROC3.2, CMAM, and GEOSCCM simulate extensive regions where seasonally or in all seasons the ozone decline exceeds 10% of the methane difference, whereas in ULAQ-CCM this effect is generally smaller than 5%. In the tropical upper-troposphere/lower stratosphere (UTLS) region, most of the models simulate a negative feedback for at least some months, i.e. methane increases cause a decrease in ozone, but the size and spatial extent of this effect is highly uncertain, with NIWA-UKCA producing ozone decreases of 10-20% of the methane difference. In most of the other models, there are some decreases, but the trends are insignificant in parts of the latitude-pressure domain at the 95% confidence level, peaking at less than 10% of the applied methane increase in CCSRNIES-MIROC3.2, CESM1-WACCM, GEOSCCM, SOCOL3, and UMSLIMCAT. CMAM exhibits no significant influence of methane on ozone in this region, and ULAQ-CCM even produces some significant increases.

The equivalent analysis for zonal-mean total-column ozone (TCO; figure 2) indicates that indeed $CH_4$ increases generally cause a TCO increase almost everywhere (apart from over the South Pole in the ULAQ-CCM). The weak responses in TCO by UMSLIMCAT and CCSRNIES-MIROC3.2 are as expected, considering the simplified treatment of tropospheric ozone in both models mentioned above. Figure S1 shows the response of ozone to methane changes, expressed in terms of ozone concentrations. From this figure, it is clear that apart from CCSRNIES-MIROC3.2 (and UMSLIMCAT, not shown) in all models the tropospheric response is a substantial albeit quite model-dependent fraction of the total-column response. In the tropics, the increase in TCO in response to $CH_4$ increases is smaller in CESM1-WACCM, CHASER-MIROC-ESM, and NIWA-UKCA than in the other models. CESM1-WACCM, CHASER-MIROC-ESM, GEOSCCM, and NIWA-UKCA also have larger TCO increases during winter/spring over the Arctic than the other models. This anticorrelation of trends in the two regions may be indicative of differences in the strength of the response of stratospheric overturning in these models, the subject of section 4.

Figure 3 shows the zonal-mean sensitivity $a$ at the surface as a function of month of the year and latitude. The seven models exhibit some common features but also some considerable qualitative and quantitative differences in their responses to methane increases. Commonalities include that methane increases cause statistically significant ozone increases everywhere. This is as expected, given the role of methane as an ozone precursor. In all seven models, the increase maximizes in northern mid-latitudes, but the seasonality of this feature varies by model. There is a secondary maximum in the Southern-Hemisphere winter. In four of the models (CESM1-WACCM, CMAM, GEOSCCM, NIWA-UKCA) the response minimizes at the South Pole during summer. CHASER-MIROC-ESM, SOCOL3, and ULAQ-CCM have a very small seasonal cycle of this feature over the South Pole. In CESM1-WACCM, there are three distinct minima in the response of ozone to methane increases, located at around 65°S in January, in the tropics throughout the year, and in the Arctic from June to September.

Differences that divide these results are partly about magnitude of the signal (NIWA-UKCA simulations show the smallest sensitivity of surface ozone to methane increases, followed roughly in order by CHASER-MIROC-ESM, CESM1-WACCM, CMAM, SOCOL3, GEOSCCM, and ULAQ-CCM). Also details of the annual cycle differ. For example, CESM1-WACCM, CMAM, and SO-COL3 produce a minimum over the Arctic in summer; there is no sign of this occurring in CHASER-MIROC-ESM, NIWA-UKCA, and ULAQ-CCM. The relatively strong response of SOCOL3 surface ozone to $CH_4$ increases may be related to a general overestimation of tropospheric ozone in the Northern Hemisphere by that model (Revell et al., 2015).

## 3.2 Sensitivity of ozone to nitrous oxide

Figure 4 shows the sensitivity to zonal-mean $N_2O$ changes ($b$) as derived from the REF-C2 and SEN-C2-fN2O experiments. The same nine models as discussed in section 3.1 also conducted SEN-

C2-fN2O. The sensitivity to $N_2O$ increases is more coherently simulated by the models than that to $CH_4$, with the models largely agreeing on the main features. In the upper stratosphere, $N_2O$ increases cause a decrease in $O_3$ of about 5 to 10 times the increase in $N_2O$, peaking in all seasons in the tropics. Above 1 hPa, there is disagreement on the sign of the ozone response, with CCSRNIES-MIROC3.2 and ULAQ-CCM producing mostly increasing ozone for increases in $N_2O$, whereas in CESM1-WACCM, GEOSCCM, NIWA-UKCA, SOCOL3, and UMSLIMCAT, the decreases dominate the increases in spatial extent. In CMAM, the co-variance of ozone with surface $N_2O$ appears to be insignificant almost everywhere above 1 hPa. In the lower stratosphere, all models produce some increases in ozone for increases in $N_2O$. This may be the result of a self-healing process, whereby ozone depletion higher up caused by increased $N_2O$ allows more UV light to penetrate to this level, producing more ozone there. The meridional extent and magnitude of the ozone increase vary by model. In CESM1-WACCM, CCSRNIES-MIROC3.2, CHASER-MIROC-ESM, GEOSCCM, NIWA-UKCA, SOCOL3, and UMSLIMCAT, the ozone increase covers the whole or almost the whole latitude range, whereas in CMAM and ULAQ-CCM the belt does not consistently extend to the poles.

Like for methane, the response of TCO to $N_2O$ changes is highly model-dependent (figure 5). (Figure S7 gives the concentration-weighted ozone responses that visualize height-dependent contributions to the TCO changes.) Best agreement in the TCO response across the nine-models ensemble is achieved in the tropics, where all models find decreases in TCO for increases in $N_2O$ ranging around $-0.075$ to $-0.05$ DU/ppbv in CCSRNIES-MIROC3.2 to roughly $-0.03$ Dobson Units (DU)/ppbv in GEOSCCM, NIWA-UKCA, SOCOL3, and ULAQ-CCM. In the northern extratropics, several of the models agree on the phasing of the annual cycle, with TCO decreases maximizing in late winter/spring and minimizing in late summer. In the southern extratropics, a similar seasonality is evident. SOCOL3 exhibits significant increases under $N_2O$ increases over Antarctica in spring (the result of large increases in ozone in the lowermost stratosphere and UTLS, figure S2), and NIWA-UKCA has relatively weak decreases and some seasonal increases under $N_2O$ increases, particularly in the Arctic in summer. Both are associated with anomalously large increases in the lower stratosphere evident in figures 4 and S2, suggesting that dynamical/chemical feedbacks in the lower stratosphere overcompensate for the additional chemical depletion that all models show in the middle stratosphere. Even for this forcing, to which the models simulate a generally consistent response in the middle stratosphere, the extratropical TCO response remains quantitatively uncertain.

Figure 6 shows $b$ evaluated at the surface. Generally, as $N_2O$ is chemically inert in the troposphere, six of the models show large areas of insignificant covariance between $N_2O$ and surface $O_3$, particularly in the extratropics. As for significant features, the same six models agree on a decrease in ozone in the tropics, also extending into northern midlatitudes in summer, of $-0.002$ to $-0.004$ times the increase in $N_2O$, and an increase of ozone by roughly $0.002$ times the increase in $N_2O$ in southern mid-latitudes during winter. In CESM1-WACCM, this feature in more pronounced, cover-

ing much of the southern extratropics, and is significant year-round. The feature is insignificant in CMAM. ULAQ-CCM, by contrast, shows significant increases in surface ozone almost everywhere for an increase in $N_2O$, peaking in northern midlatitudes, i.e. it is in disagreement with the other models regarding both magnitude and shape of the annual cycle of $b$.

### 3.3 Sensitivity of ozone to equivalent chlorine

Figure 7 shows the sensitivity of zonal-mean ozone to changes in $Cl^{eq}$ (section 1), as derived from the REF-C2 and SEN-C2-fODS experiments. Eight models have conducted both of these experiments. In the upper stratosphere, there is a significant decrease in ozone by up to 300 to 1000 times the $Cl^{eq}$ increase. This is consistently simulated by all models, and is the consequence of global halogen-catalyzed ozone depletion maximizing at around 1 to 10 hPa. Higher up, above approximately 1 hPa, the models simulate mostly a decrease of 0 to 50 times the $Cl^{eq}$ increase. There also are consistent decreases in ozone in the lower stratosphere / tropopause region of the southern high latitudes during spring and summer, associated with the Antarctic ozone hole. In January, in what is likely a dynamical feedback, there is an increase in ozone (for an increase in ODSs) between about 50 and 10 hPa. In CCSRNIES-MIROC3.2, CESM1-WACCM, CHASER-MIROC-ESM, CMAM, and UMSLIMCAT, Antarctic October polar ozone depletion occupies the entire lower stratosphere, between $\sim 200$ and 10 hPa, with ozone loss reaching 1000 times the difference in $Cl^{eq}$.

Regarding the response of the TCO to $Cl^{eq}$ changes, the models uniformly exhibit decreases in TCO for an increase in $Cl^{eq}$ (figure 8). In the tropics, there is reasonable agreement regarding the size of the effect. In the extratropics, there is some quantitative disagreement. Best agreement is found over the Antarctic in spring, where most models in October agree to within $\pm 10$ DU/ppbv($Cl^{eq}$) with each other. This general agreement may be the result of a long-term focus on this region for the impact of ozone depletion. By contrast, in the Arctic significant quantitative differences are apparent regarding this effect, also evident in figure S3. In all models except ACCESS-CCM and NIWA-UKCA, the reduction of TCO in the Arctic is significantly weaker than in the Antarctic.

As for surface ozone, there is little agreement as to the impacts of this stratospheric ozone depletion (figure 9). In ACCESS-CCM and NIWA-UKCA, there is a widespread decrease in surface ozone associated with stratospheric ozone depletion, with maxima in both mid-latitude regions during autumn. The southern one is larger, reaching the size of the difference in $Cl^{eq}$. The near-symmetry between the two hemispheres is in agreement with the pronouced Arctic ozone depletion produced by ACCESS-CCM and NIWA-UKCA (figure 7). CESM1-WACCM and CMAM produce a Southern-Hemisphere maximum of similar magnitude, but CMAM produces a secondary maximum over the South Pole in austral spring, and the response in the Northern Hemisphere in both models is much smaller than in ACCESS-CCM and NIWA-UKCA. CHASER-MIROC-ESM shows a much weaker response to $Cl^{eq}$ and also only minor asymmetries between the hemispheres. ULAQ-CCM disagrees with the other five models in that in the Northern Hemisphere and the tropics, ozone mostly increases

under increases of $Cl^{eq}$. In the southern extratropics, this model largely produces decreases but the effect maximizes in austral summer, i.e. the seasonality disagrees with the other five models.

It is noteworthy that four of the six models display their peak response of surface ozone to stratospheric ozone depletion in austral autumn, approximately 6 months after the onset of the Antarctic ozone hole.

**3.4 Sensitivity of ozone to GHGs**

Here we assess the sensitivity of ozone to increases in $CO_2^e$ (section 1). Increases in $CO_2^e$ cause increases of ozone peaking between roughly 10 and 1 hPa; these increases are of similar magnitude in all models (figure 10). They also cause decreases in ozone in the tropical and subtropical lower stratosphere; again there largely is agreement about the magnitude of this effect. Both the

350 decrease and the increase may be aspects of an upward displacement and associated acceleration of the Brewer-Dobson Circulation (Butchart, 2014; Oberländer-Hayn et al., 2016, ; section 4). Also stratospheric cooling, through its impact on ozone-depleting chemical cycles, leads to an increase in stratospheric ozone. In the mesosphere, there is quantitative disagreement regarding the impact of increases in $CO_2^e$. CESM1-WACCM, CMAM, ULAQ-CCM, and UMSLIMCAT exhibit mostly or

355 generally increases, whereas in NIWA-UKCA and CCSRNIES-MIROC3.2 increases cause ozone to decline. The models generally agree on a region of ozone decrease in the tropical and subtropical lower stratosphere which reaches $-0.5 \times 10^{-3}$ to $-2 \times 10^{-3}$ times the increase in the $CO_2^e$ VMR.

Regarding the TCO response to $CO_2^e$ increases (figure 11), there is reasonable agreement across the models. In all models, there is significant cancellation in the tropics between decreases in ozone

in the lower stratosphere with increases in the middle and upper stratosphere and (for some models) in the troposphere (figure S4). In five of the models (CCSRNIES-MIROC3.2, CHASER-MIROC-ESM, CMAM, NIWA-UKCA, and ULAQ) this tropical TCO decreases under increasing $CO_2^e$ (Eyring et al., 2010), whereas in two (CESM1-WACCM, UMSLIMCAT) it increases. In order to assess whether for CESM1-WACCM the finding is the result of the linear analysis conducted here, whose

limitation is that nonlinear interactions between increases of $CO_2^e$, $N_2O$, and $CH_4$ are ignored, we analyze here a simulation using CESM1-WACCM in which is identical to the REF-C2 simulations except that $CO_2$ is held fixed at 1960 levels. In this simulation, actually we find that CESM1-WACCM does produce a small decrease of tropical TCO for increasing $CO_2$ in much of the tropics, much of the time (figure 12). This decrease is still smaller than in most other models, but the finding

does indicate that the tropical ozone feedback is subject to substantial nonlinear coupling between the forcings which we cannot fully diagnose here. Also UMSLIMCAT produces increases of tropical TCO for increasing $CO_2^e$; we attribute this partly to the prescribed tropospheric ozone in this model. Increases in the Northern extratropics during boreal winter and spring are consistent across the seven models; they exceed those in the South. There is no agreement regarding the seasonality

of the effect in the southern extratropics. CHASER-MIROC-ESM, CMAM, and UMSLIMCAT pro-

duce some significant decreases in TCO in response to $CO_2^e$ increases over the South Pole in austral winter and/or spring; the other models do not simulate this feature.

As for surface ozone, CMAM, CHASER-MIROC-ESM, NIWA-UKCA, and ULAQ-CCM mostly produce decreases of surface ozone for an increase in $CO_2^e$, but also some increases at northern high latitudes during autumn, winter, and spring (figure 13). CESM1-WACCM produces smaller changes in ozone under climate change; they are negative (0 to $-5$ ppbv/ppmv) in the tropics and in the SH during summer, also in the Arctic from late spring to autumn and positive (0 to 5 ppbv/ppmv) at other times and seasons. In ULAQ-CCM, increases are restricted to late winter and spring in the Arctic and to October in the Antarctic. While the models agree about decreases in ozone in the tropics and mid-latitudes, there is disagreement about the magnitude, with decreases in CESM1-WACCM and NIWA-UKCA smaller than in the other models. CESM1-WACCM, CHASER-MIROC-ESM, and NIWA-UKCA simulate relatively large ozone decreases over the Arctic in summer. These may be the result of reductions of sea ice cover and associated decreased tropospheric ozone formation in an ice-albedo feedback on photochemistry (Voulgarakis et al., 2009). Note that three of the model used here (CESM1-WACCM, CHASER-MIROC-ESM, and NIWA-UKCA) are coupled atmosphere-ocean models, but this has no direct bearing on this ice-albedo feedback because the other models use prescribed ocean-surface fields that also have sea ice generally decreasing in spatial extent as global warming progresses (Morgenstern et al., 2017).

## 4   What is causing the differences in the responses of ozone?

In the previous sections, we have shown that the responses of total-column, lower-stratospheric, and surface ozone to the anthropogenic forcings studied here vary considerably by model. By contrast, in the middle and upper stratosphere, we find a more consistent response. This indicates that broadly speaking, gas-phase chemistry schemes appear to be relatively consistent across the model ensemble studied here, but dynamical feedbacks (that influence the responses in the lower-stratosphere) are not. In this context we assess how stratospheric age of air (AOA) responds to these forcings (for a review of AOA see Waugh and Hall, 2002). AOA is the average time it takes an air parcel to travel from the troposphere to any given location in the stratosphere. It is a measure of the strength of the Brewer-Dobson Circulation (BDC). Essentially, we explore the hypothesis that differences in the response of the BDC to anthropogenic forcings are behind some of the differences in the response of ozone to these forcings. Hence we repeat the analysis formulated in section 2.2.1 but now replacing ozone with AOA. Of the ten models considered here, six have produced sufficient output for this, i.e. AOA from the REF-C2 and at least one of the sensitivity simulations. These models are ACCESS-CCM, CCSRNIES-MIROC3.2, CESM1-WACCM, CMAM, NIWA-UKCA, and ULAQ-CCM. Of these models, ACCESS-CCM, CCSRNIES-MIROC3.2, CMAM, and ULAQ-CCM use prescribed sea surface forcing, with identical forcing used for REF-C2 and the SEN-C2

simulations. This restricts the climate response particularly in the troposphere to the variant forcings explored in the SEN-C2 simulations.

In summary, we find the following: (The figures discussed here are in the supplement.)

– Increases of $N_2O$ in REF-C2 produce mostly insignificant differences in AOA in all five models considered here, versus the corresponding SEN-C2-fN2O simulations (figure S5). This suggests that the impact of $N_2O$ changes on ozone is caused mostly directly by chemistry, with only a minor role for dynamical feedbacks. Speculatively, such a minor role for dynamics might be the result of a cancellation of the impacts on stratospheric dynamics of the radiative forcing exerted by $N_2O$ increases with those due to ozone depletion associated with such increases. Such a cancellation would mean that dynamical feedbacks do not interfere much with the relatively good agreement in the chemical model responses to $N_2O$ increases discussed in section 3.2, which results from the similar gas-phase chemistry schemes employed by the models. However, the CMAM SEN-C2-fN2O did not use the reduced $N_2O$ in the radiation scheme (for radiation, $N_2O$ in this model follows the same scenario as in REF-C2). CMAM still exhibits a near-zero impact of reduced $N_2O$ on AOA, suggesting that this mechanism may not hold for all models.

– Increases in $CH_4$ lead to significant reductions in AOA above roughly 100 hPa in CESM1-WACCM and NIWA-UKCA, weaker or insignificant changes in CCSRNIES-MIROC3.2 and CMAM, and some increases in age in much of the stratosphere in ULAQ-CCM (figure S6). This behaviour corroborates figure 2 where CESM1-WACCM and NIWA-UKCA show relatively small sensitivities of tropical column ozone to increases in $CH_4$ and large sensitivities of springtime Arctic ozone, suggesting that in these models the speed-up of the BDC accompanying $CH_4$ increases contributes to the sensitivity of TCO to $CH_4$ increases. Such a speed-up removes ozone from the tropics and transports it to the winter/spring pole, contributing to this contrast in sensitivity. By contrast, CMAM and ULAQ-CCM are characterized by a relatively weak contrast in the trend in AOA between the tropics and the polar latitudes, consistent with their response in AOA to increasing $CH_4$ (figure 2). In the case of CMAM, this may be because in this model, actually the reduced $CH_4$ characterizing the SEN-C2-fCH4 experiment was only used in chemistry but not in radiation. The radiation scheme saw a similar $CH_4$ evolution as in the REF-C2 simulations. Hence only differences in ozone have affected the AOA response in this model.

An additional analysis of the temperature response to $CH_4$ increases (not shown) indicates that the models also exhibit considerable variations in their temperature trends in response to methane changes. Most indicate stratospheric cooling of varying magnitude but some also warming of the stratosphere. This might begin to explain the differences in age-of-air.

– Increases in $Cl^{eq}$ lead to significant and similar decreases in age throughout most of the stratosphere in five of the models but not in CCSRNIES-MIROC3.2; this model produces mostly no significant change in response to this forcing (figure S7). The only region that shows consistent increases in age is the Antarctic polar vortex which in all models shows increasing AOA during summer, suggesting an increasing persistence into summer. A comparison with figure 7 indicates that the region of increasing age during January coincides with the region of ozone depletion at the base of the polar vortex. Of the five models considered here, CCSRNIES-MIROC3.2 has the largest difference in sensitivity between tropical and Antarctic springtime total-column ozone (figure 8), which is consistent with the lack of speed-up of the BDC in this model, compared to the other five. The role of ozone depletion in driving a decrease in AOA, shown by most of the models analyzed here, has been found before (e.g. Polvani et al., 2017).

In ACCESS-CCM and NIWA-UKCA, the region of increasing age for increasing $Cl^{eq}$ in January is located somewhat higher in the atmosphere than in the other models. This has been noted before, in the context of the evaluation of ozone depletion in the ACCESS-CCM (Stone et al., 2016). (Note again ACCESS-CCM and NIWA-UKCA share the same atmosphere model.)

– Increases in $CO_2^{eq}$ cause consistent decreases of AOA above about 100 hPa in all five models shown here, with CMAM and CCSRNIES-MIROC3.2 exhibiting a larger response than CESM1-WACCM, NIWA-UKCA, and ULAQ-CCM (figure S8). Below 100 hPa, all models show decreases in age in the extratropical lowermost stratosphere, except for CCSRNIES-MIROC3.2 which also shows some significant and substantial increases in age around the 100 hPa pressure level. CESM1-WACCM, CMAM, NIWA-UKCA, and ULAQ-CCM exhibit a region of weak increases of age, or insignificant sensitivity of age, in response to increasing $CO_2^{eq}$, in the tropical upper troposphere. In CMAM, NIWA-UKCA, and ULAQ-CCM, this "tongue" extends roughly 200 hPa, but in CESM1-WACCM it extends significantly above the tropical tropopause, to about 80 to 100 hPa. This difference in behaviour is a contributing factor in the weak response of tropical TCO in CESM1-WACCM to increasing $CO_2^{eq}$. Conversely, the large difference in sensitivity of TCO in CMAM between the tropics and the extratropics is related to the relatively large speed-up of the BDC in response to $CO_2^{eq}$ forcing in this model.

These considerations do not constitute a complete discussion of the differences in model behaviour found in this paper. But they do corroborate the hypothesis that dynamics and transport contribute to the sensitivity of modelled ozone to the anthropogenic forcings considered here. Some interesting inconsistencies in model behaviour are found here that require further analysis.

## 5 Linearity of the ozone response to greenhouse gas forcing

Based on the previous sections, we calculate, assuming linear scaling and ignoring non-linear coupling (Portmann et al., 2012; Dhomse et al., 2016), the ozone fields that would result from GHG scenarios other than the RCP 6.0 forcing used in REF-C2. For the moderate-emissions scenarios RCP 2.6 and 4.5, this can be seen as a consistency test. For the more extreme RCP 8.5, where forcings are partially outside the range spanned by RCP 6.0 / REF-C2 and the total ozone abundance is larger than in REF-C2, this exercise will help highlight nonlinear couplings between the forcings. The scaling is possible for those models that have produced the REF-C2, SEN-C2-fGHG, SEN-C2-fN2O, and SEN-C2-fCH4 simulations. We produce scaled ozone fields for CCSRNIES-MIROC3.2, CESM1-WACCM, CMAM, ULAQ-CCM, and UMSLIMCAT ( CHASER-MIROC-ESM and NIWA-UKCA did not produce any SEN-C2-RCP simulations needed for comparison here). For the more moderate RCPs 2.6 and 4.5, the ozone fields resulting from such scaling in the zonal mean relatively accurately match those simulated by the five models. Significant relative differences occur in the troposphere, where the scaling method is not applicable (see above) and in the UTLS region, where changes in the tropopause height constitute a non-linear feedback not well captured by simple scaling of the ozone fields (supplement, figures S9 and S10). Larger differences, generally of opposite sign relative to RCP2.6 and RCP 4.5, occur for RCP 8.5. Here, the models fall into two groups: One group, comprising CCSRNIES-MIROC3.2, CESM1-WACCM, and CMAM, overestimate ozone in this scaling in the mid- and upper stratosphere and underestimate it in the mesosphere (above 1 hPa). A second group, comprising ULAQ-CCM and UMSLIMCAT, overestimates ozone almost everywhere above the UTLS region, ULAQ-CCM more so than UMSLIMCAT. In all cases, the analysis quantifies that nonlinear interactions play a significant role, particularly in the RCP8.5 scenario.

## 6 Some general thoughts on the generation of a consensus ozone database

As noted in section 1, the CMIP6 activity requires prescribed ozone fields to drive simulations by CMIP6 models that do not interactively compute ozone. Out of twenty models participating in CCMI-1, only two were actually used in the generation of the ozone climatology provide to CMIP6 participants, namely CMAM and CESM1-WACCM (M. Hegglin, personal communication). Such a narrow base was chosen because these two modelling groups were ready to provide pre-industrial and pre-1960 ozone fields that are also required for CMIP6 but fall outside the period spanned by CCMI-1 simulations. A larger and more representative base of model simulations might have been possible to use here, had the production of CMIP6 ozone climatologies been identified early on as a key deliverable of the CCMI-1 activity, particularly in view of the several coupled atmosphere-ocean CCMs participating in CCMI-1 that would have had to conduct spin-up simulations covering the pre-1960 period.

It is not the purpose of the present paper to actually produce such a merged ozone climatology. Nevertheless, we offer some thoughts on how one might go about producing such a climatology.

1. All ozone fields are interpolated to a common pressure-based grid, as is a reference ozone climatology derived from satellite data and in-situ observations. Single-model ensemble means are formed for those models that have produced more than one ensemble member.

2. It is clear that not every model is equally suitable for representing ozone in every region. For example, some models have prescribed ozone in the troposphere or do not extend into the mesosphere. This can be accounted for introducing, for every model $i$, weighting functions $\zeta_i(p)$ that are zero outside the pressure interval where model $i$ should be considered. Also the weights can include information on ensemble size. This accounts for the idea that the statistical uncertainty in model projections reduces with increasing ensemble size. In addition to such elementary considerations, it is possible to give models weights based on skill scores, but these depend on metrics chosen to measure skill, which can be contentious.

3. The multi-model mean is formed, using the above weights:

$$\overline{O_3} = \frac{\sum \zeta_i O_3^i}{\sum \zeta_i}. \tag{4}$$

4. Forming a multi-model mean already has the effect of dampening interannual variations. These can be further reduced by applying a filter.

5. Bias-correcting the ozone fields versus observational ozone climatologies is possible. However here a few caveats apply: (a) Available ozone climatologies have their own shortcomings, particularly in the troposphere where space-borne measurements are difficult or subject to large uncertainty. (b) In the stratosphere, and to some extent in the troposphere, the dependence of ozone on variations in long-lived constituents can be expressed in terms of a regression model. Using a modelling approach, it is possible, as demonstrated here, to identify the contributions made by individual long-lived gases to long-term ozone trends. However, the satellite record may not be straightforwardly amenable to such an approach because multiple forcings are acting simultaneously whose effects likely cannot be separated using multi-variate regression – the record may be too short, meteorological noise too large, or impacts of different forcings too similar for this to be a viable strategy. This means only a simpler approach may be possible, consisting of subtracting the bias in the mean annual cycle of ozone, determined for the satellite era, off the multi-model mean. The problem here is that the bias may be a function of the anthropogenic forcings. If that is the case, simply subtracting off the mean bias could result in inappropriate "corrections", particularly before and after the satellite era.

6. Unlike previous CMIP rounds, for CMIP6 zonally resolved ozone will be requested. Stratospheric ozone is subject to zonal asymmetries caused by dynamical anomalies e.g. due to

orographic forcing. For example, there is a significant trend in the orientation of the Antarctic polar vortex during the satellite era which some models fail to reproduce (Dennison et al., 2017). Given the inability to attribute such misbehaviour to individual anthropogenic forcings as discussed above, it appears difficult though to consistently account for this in a correction.

With these considerations in mind, apart from the restricted database, taking a simple weighted average of available modelled ozone fields (M. Hegglin, personal communication) appears to be the most practical and straightforward approach to the problem. In comparison to the process adopted for CMIP5 ozone (Cionni et al., 2011), for CMIP6 there will not be any discontinuity between stratospheric and tropospheric ozone, and the ozone climatology now will be zonally resolved everywhere.

## 7   Conclusions

We have analysed the sensitivities of ozone to changes in $CH_4$, $N_2O$, halogenated ODSs, and a combination of $CO_2$ and other greenhouse gases in ten CCMI-1 models. In all cases we find some qualitative and quantitative agreement, mainly about the impacts in the middle stratosphere, but also considerable disagreements in other regions, particularly the troposphere, the UTLS region, and the mesosphere. The middle-stratospheric impact of $CH_4$ increases is largely consistently simulated by the nine models studied here, but significant differences occur in the lower stratosphere, the troposphere, and in the total-column impacts of increasing $CH_4$. The impacts on ozone of increasing $N_2O$ are relatively consistently simulated, in particular regarding decreases in the middle stratosphere and increases in the lower stratosphere. Also six of the models agree to some extent on the relatively small impact on surface ozone. However, as with $CH_4$, quantitative differences in the sensitivity of lower-stratospheric ozone to increases of $N_2O$ mean that the response of the TCO to $N_2O$ increases remains uncertain. The impact of changing ODSs on stratospheric ozone is well simulated, with some general agreement regarding the middle-stratospheric response and also the impact on polar ozone. There remain quantitative differences regarding the impact on the TCO, globally, and particularly regarding the impact of stratospheric ozone depletion on surface ozone. Lastly, we have studied the effect of a combination of $CO_2$ and other GHGs on ozone. Essentially, global warming causes ozone in the middle stratosphere to increase and in the low-latitude lower stratosphere to decrease. The TCO impacts are relatively consistently simulated, but the response of surface ozone to global warming remains highly uncertain, with the five CCMI-1 models suitable for this analysis disagreeing on major aspects of the impact. They exhibit larger differences regarding the impact of global warming on surface ozone than were found in a recent study using a different ensemble (Young et al., 2013). This may reflect uncertainties related to stratosphere-troposphere coupling that were suppressed in the large subset of the models examined by Young et al. (2013) which used prescribed stratospheric ozone. This may thus be an example of additional model complexity causing increased divergence of results (Morgenstern et al., 2017).

In an effort to further investigate the dynamical feedbacks causing some differences in model response to these anthropogenic feedbacks, we have analyzed AOA in a subset of the models studied here. Here we find some distinct consistencies and inconsistencies in the response of AOA to these forcings. With further analysis, the results might help shed light on the actual causes of these inter-model variations. Considering that greenhouse gases interact with dynamics via their impact on radiation, the consistency of the impact of greenhouse gases on radiative heating might be worth assessing in more detail.

In essence, it appears that mid- and upper-stratospheric impacts of the four gaseous anthropogenic forcings are relatively consistently simulated by the subset of CCMI-1 models studied here, but lower-stratospheric, tropospheric, and mesospheric impacts often are not. The total-column response is affected by dynamical feedbacks which are not consistent in the CCMI-1 model ensemble. We have linked these to differences in the impact on stratospheric overturning. These inconsistencies in the CCMI-1 ensemble need to be considered and may have consequences for the fidelity of any merged ozone climatologies produced from the CCMI-1 results.

It is possible that the results presented here are subject to a sampling bias in the sense that they require a relatively large number of sensitivity simulations to be available, which some more expensive, higher-resolution models in the CCMI-1 ensemble have not performed. It is regrettable that even though the CCMI-1 ensemble nominally comprises 20 models (Morgenstern et al., 2017), only ten models have been considered here, and of these, some are unsuitable for certain diagnoses, e.g. because tropospheric composition is prescribed or because required simulations or diagnostics do not exist. Nonetheless, the results point to the need to better characterize quantitatively the lower-stratospheric climate-ozone feedbacks that are the likely cause for the discrepancies found here. The impact of methane on ozone occurs significantly in the troposphere. Here differences in formulation and sophistication of tropospheric chemistry also impact the models' responses to methane changes. Such differences may also play into the responses to the other forcings, although the surface ozone responses to $N_2O$ increases are surprisingly consistent across most of the models, despite such differences in formulation.

## 8   Availability of simulations

The ozone fields as used here are mostly as downloaded from the Centre for Environmental Data Analysis (CEDA; ftp://ftp.ceda.ac.uk). CESM1-WACCM data have been downloaded from http://www.earthsystemgrid.org. For instructions for access to both archives see http://blogs.reading.ac.uk/ccmi/badc-data-access. Some data have also been supplied directly by the co-authors; these data will in due course be uploaded to the CEDA archive.

## Appendix A: Calculation of significance intervals

In the calculation of the regression coefficients $a$, $b$, $c$, and $d$ of equation 1 confidence intervals are critical for understanding where the regression coefficients differ from 0, i.e. where the uncertainty in them exceeds the amplitude. For this a standard statistical approach is used which essentially assumes that the residual $\epsilon$ consists of "white noise", i.e. there is no autocorrelation.

For this we use an IDL routine "trend.pro" (D. Stone, personal communication). The regression coefficients simply come out of a least-squares regression which uses the difference timeseries in ozone versus the various external forcing (section 2.2.1).

Given are the original time series $y$ of simulated ozone differences at a given latitude, pressure level, and month of the year, $n$ years in length, and the associated external forcing $x$ (such as an annual global-mean methane mixing ratio). Then let $y_{fit}$ be the vector of best-fit regression values. Next we define

$$s_e = \sqrt{\frac{\sum \epsilon^2}{n-2}} \tag{A1}$$

and

$$s_{xx} = \sqrt{\sum \left(x - \overline{x}\right)^2} \tag{A2}$$

where $x$ represents one of the four forcings considered here. We calculate the confidence interval $\kappa$ that characterizes the distribution:

$$\kappa = t_{cvf}(0.025, n-2)\frac{s_e}{s_{xx}} \tag{A3}$$

Here, $t_{cvf}$ is the cut-off value of Student's $t$ distribution with $n-2$ degrees of freedom. The numerical value 0.025 means that $\kappa$ refers to the 95% confidence interval.

More details on this process are in the routine used here (http://web.csag.uct.ac.za/~daithi/idl_lib/pro/trend.pro) and in the documentation of the $t_{cvf}$ function (e.g., http://northstar-www.dartmouth.edu/doc/idl/html_6.2/T_CVF.html).

For the above approach to be robust, the residual $\epsilon$ (equation 1) needs to be free of auto-correlation. We test this using the Durbin-Watson criterion (Durbin and Watson, 1950; Morgenstern et al., 2014):

$$d = \frac{\sum_{i=2}^{n} \left(\epsilon_i - \epsilon_{i-1}\right)^2}{\sum_{i=1}^{n} \epsilon_i^2} \tag{A4}$$

In all situations $0 \leq d \leq 4$. $d = 2$ would characterize a dataset without autocorrelation. For $n = 140$ or 141, the case considered here, and at 95% confidence,

$$1.6 \leq d \leq 2.4 \tag{A5}$$

would characterize a dataset very likely free of autocorrelation (https://www3.nd.edu/~wevans1/econ30331/Durbin_Watson_tables.pdf). In figures S1-S4, violations of the Durbin-Watson citerion

are marked with stippling. Autocorrelation does indeed play a role in all models, diagnostics, and seasons, but to varying extents. In principle, autocorrelation can have two different origins, namely genuine modes of variability that operate on scales of a year or longer, e.g. the Quasi-Biennial Oscillation, or alternatively nonlinear aspects to the response of the model to the forcings, which might mean that the linear regression fit systematically over- or underpredicts the model behaviour for extended periods of time. The first cause would recede with increasing ensemble size, the second might increase relative to the random noise that is suppressed by increasing ensemble sizes. The figures S1-S4 indicate that the models with larger ensemble sizes are equally or more affected by autocorrelation than those with small ensemble sizes, suggesting that non-linearities may well play a role in this. However, a more in-depth analysis of this aspect is needed.

*Acknowledgements.* We thank the Centre for Environmental Data Analysis (CEDA) for hosting the CCMI-1 data archive. We acknowledge the modelling groups for making their simulations available for this analysis, and the joint WCRP SPARC/IGAC Chemistry-Climate Model Initiative (CCMI) for organizing and coordinating this model data analysis activity. We acknowledge the UK Met Office for use of the MetUM. This research was supported by the NZ Government's Strategic Science Investment Fund (SSIF) through the NIWA programme CACV. OM acknowledges funding by the New Zealand Royal Society Marsden Fund (grant 12-NIW-006) and by the Deep South National Science Challenge (http://www.deepsouthchallenge.co.nz). The authors wish to acknowledge the contribution of NeSI high-performance computing facilities to the results of this research. New Zealand's national facilities are provided by the New Zealand eScience Infrastructure (NeSI) and funded jointly by NeSI's collaborator institutions and through the Ministry of Business, Innovation & Employment's Research Infrastructure programme (https://www.nesi.org.nz). ACCESS-CCM runs were conducted under the Australian Antarctic Science Project 4012 (FORCeS). WACCM is a component of NCAR's Community Earth System Model (CESM), which is supported by the National Science Foundation (NSF). Computing resources (ark:/85065/d7wd3xhc) were provided by the Climate Simulation Laboratory at NCAR's Computational and Information Systems Laboratory, sponsored by the National Science Foundation and other agencies. The SOCOL team acknowledges support from the Swiss National Science Foundation under grant agreement CRSII2_147659 (FUPSOL II). CCSRNIES's research was supported by the Environment Research and Technology Development Fund (2-1303 and 2-1709) of the Ministry of the Environment, Japan, and computations were performed on NEC-SX9/A(ECO) and NEC SX-ACE computers at the CGER, NIES.

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

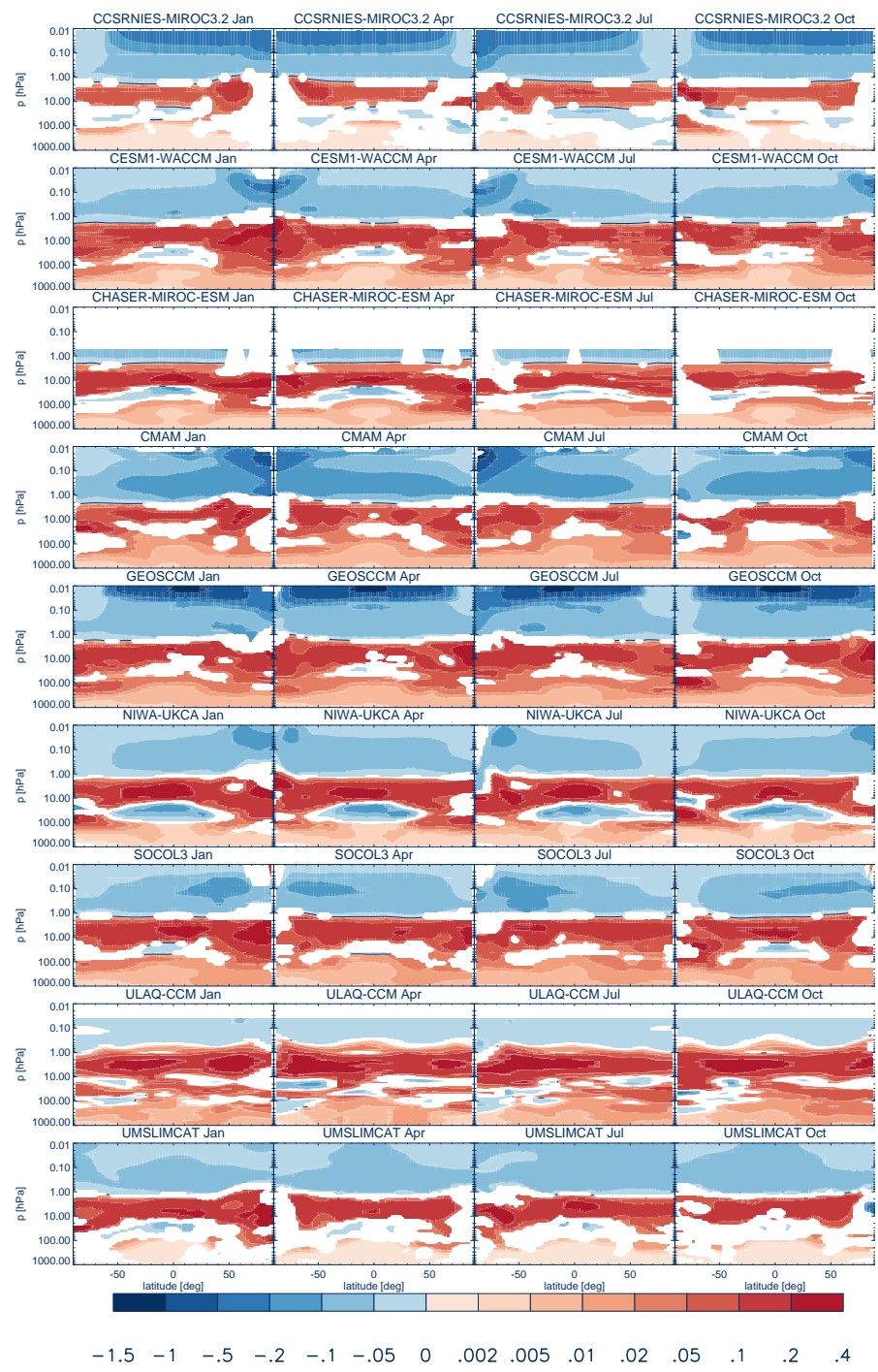

Figure 1: Ratio of zonal-mean ozone volume mixing ratio changes to VMR changes in surface $CH_4$ ($a$) as derived from the REF-C2 and SEN-C2-fCH4 simulations. $a$ is dimensionless. The colour white indicates that $a$ is not significantly different from 0 at the 95% confidence interval. The plots for ULAQ-CCM (bottom row) have no data above 0.04 hPa.

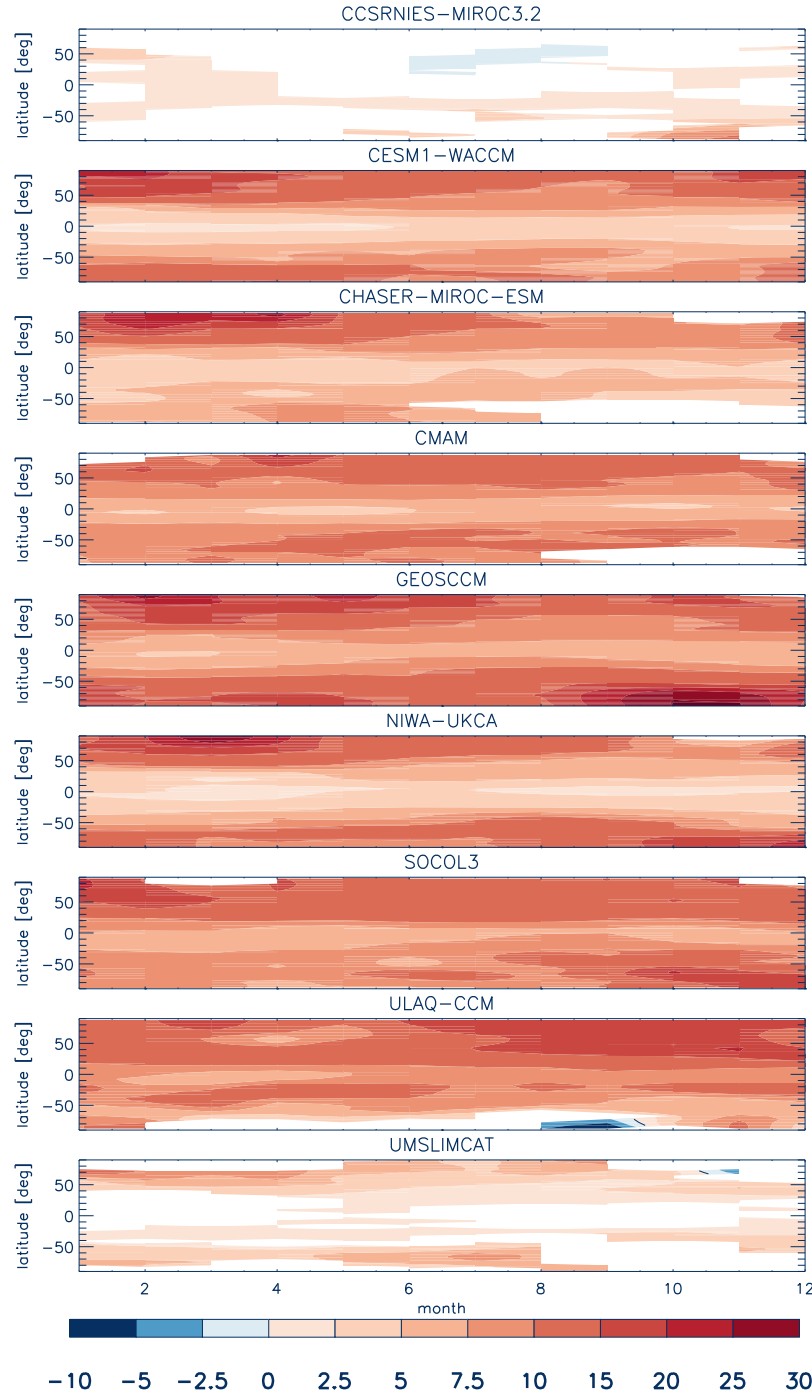

Figure 2: Ratio of zonal-mean total-column ozone changes to VMR changes in surface $CH_4$ (in Dobson Units / ppmv) as derived from the REF-C2 and SEN-C2-fCH4 simulations. The colour white indicates insignificantly differences from 0 at the 95% confidence interval.

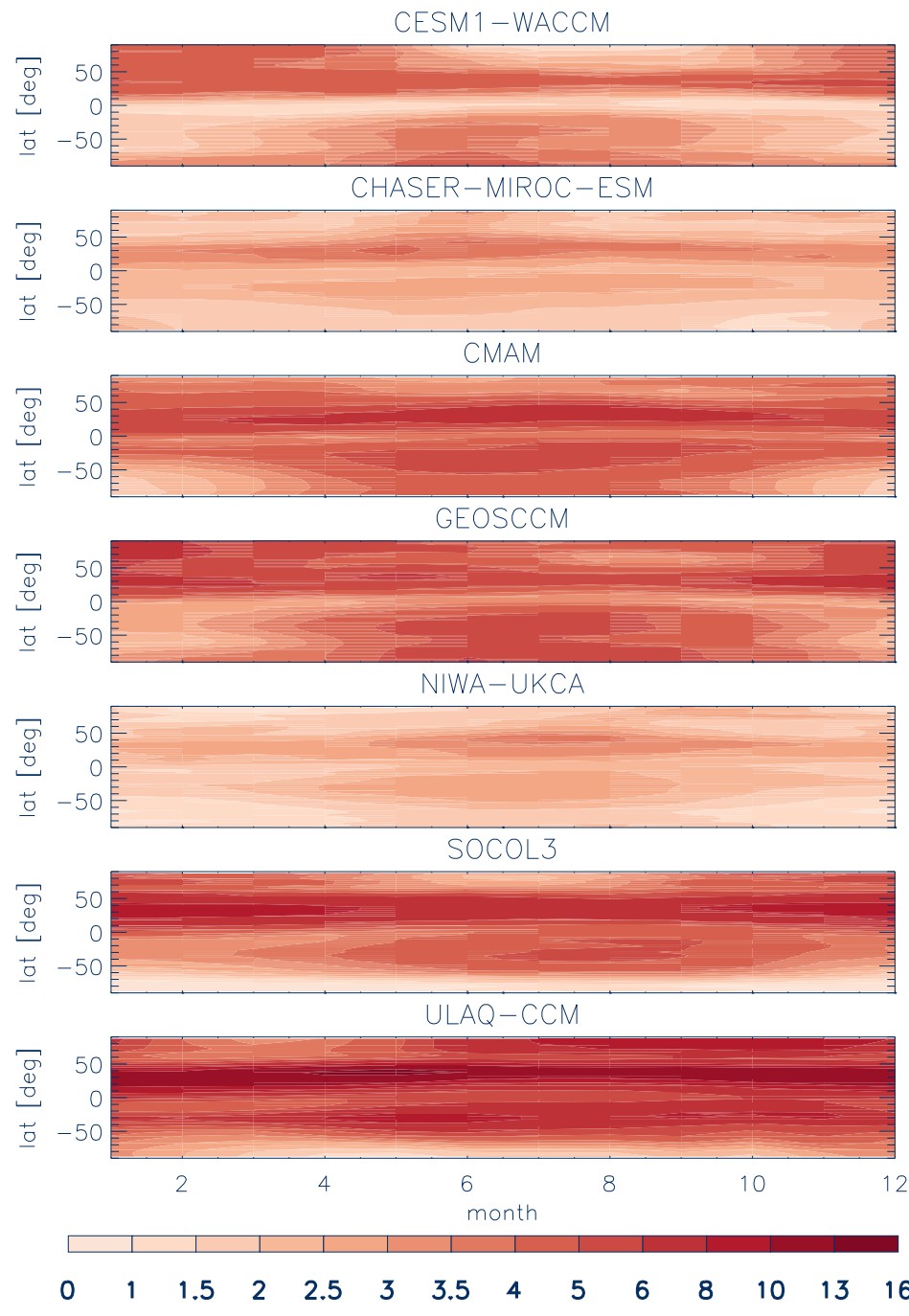

Figure 3: Ratio of zonal-mean surface ozone changes to to changes in surface $CH_4$ (in ppbv / ppmv) as derived from the REF-C2 and SEN-C2-fCH4 simulations.

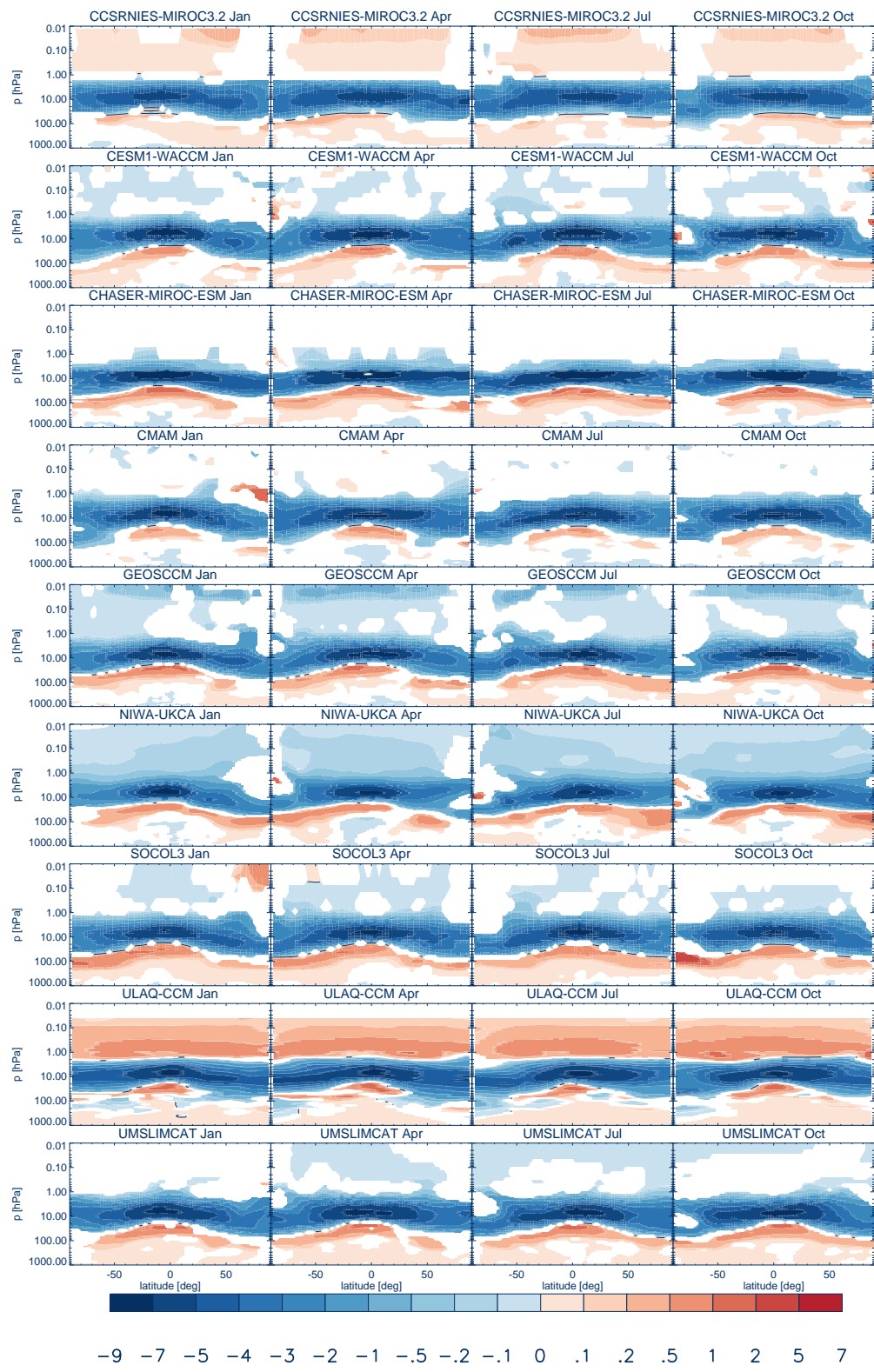

Figure 4: Same as figure 1 but for $N_2O$.

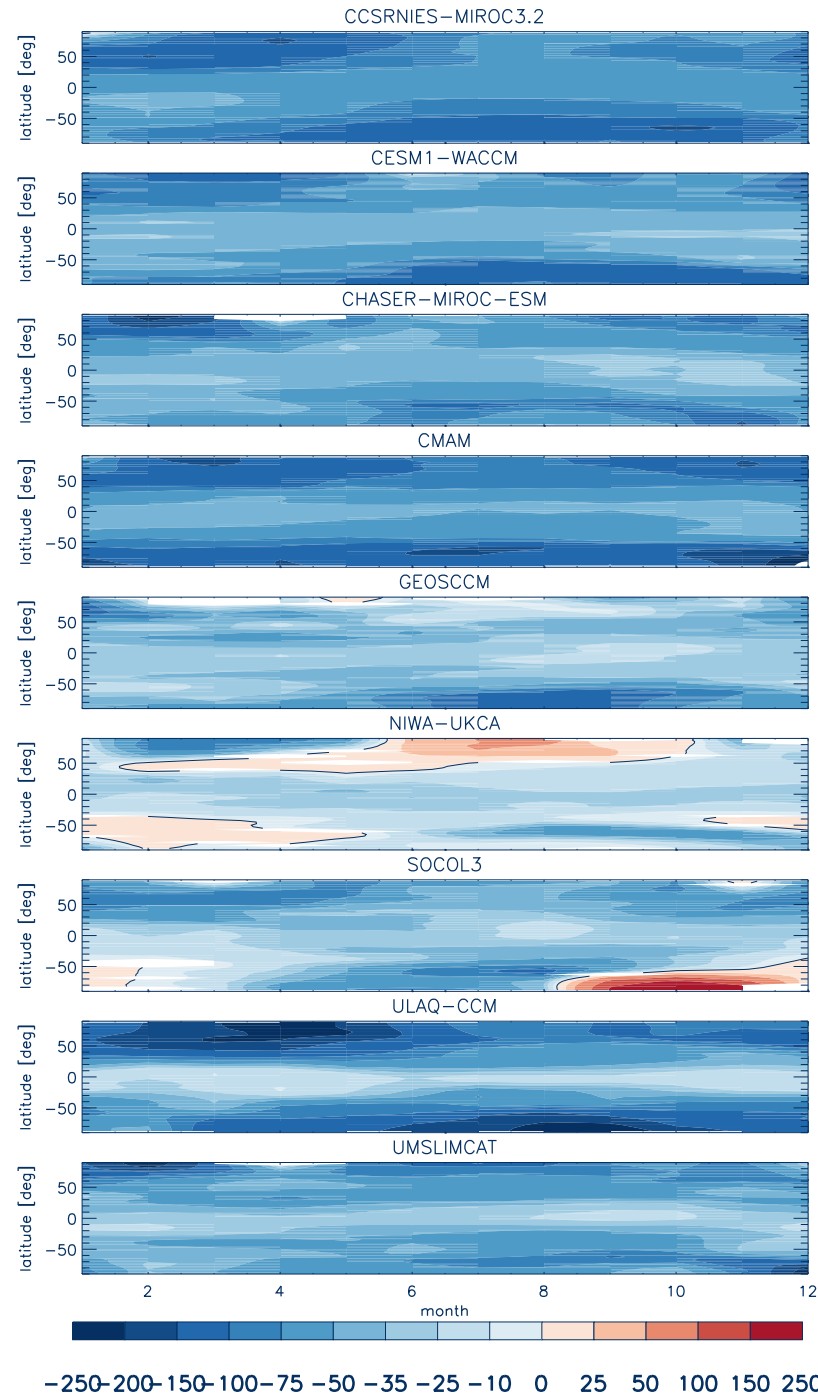

Figure 5: Same as figure 2 but for $N_2O$, in units of DU/ppmv, derived from the REF-C2 and SEN-C2-fN2O simulations.

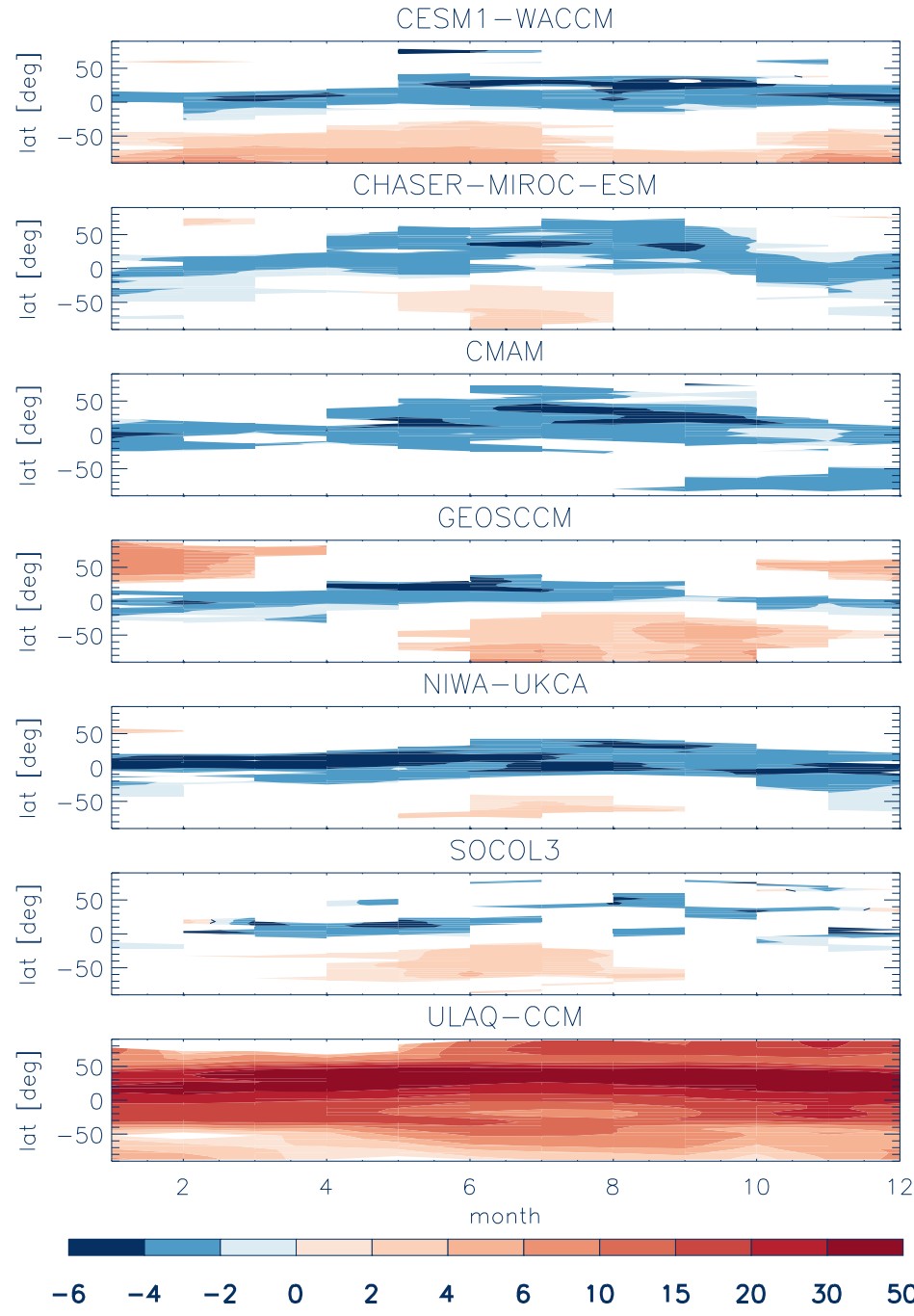

Figure 6: Same as figure 3 but for $N_2O$, in ppbv/ppmv, as derived from the REF-C2 and SEN-C2-fN2O simulations.

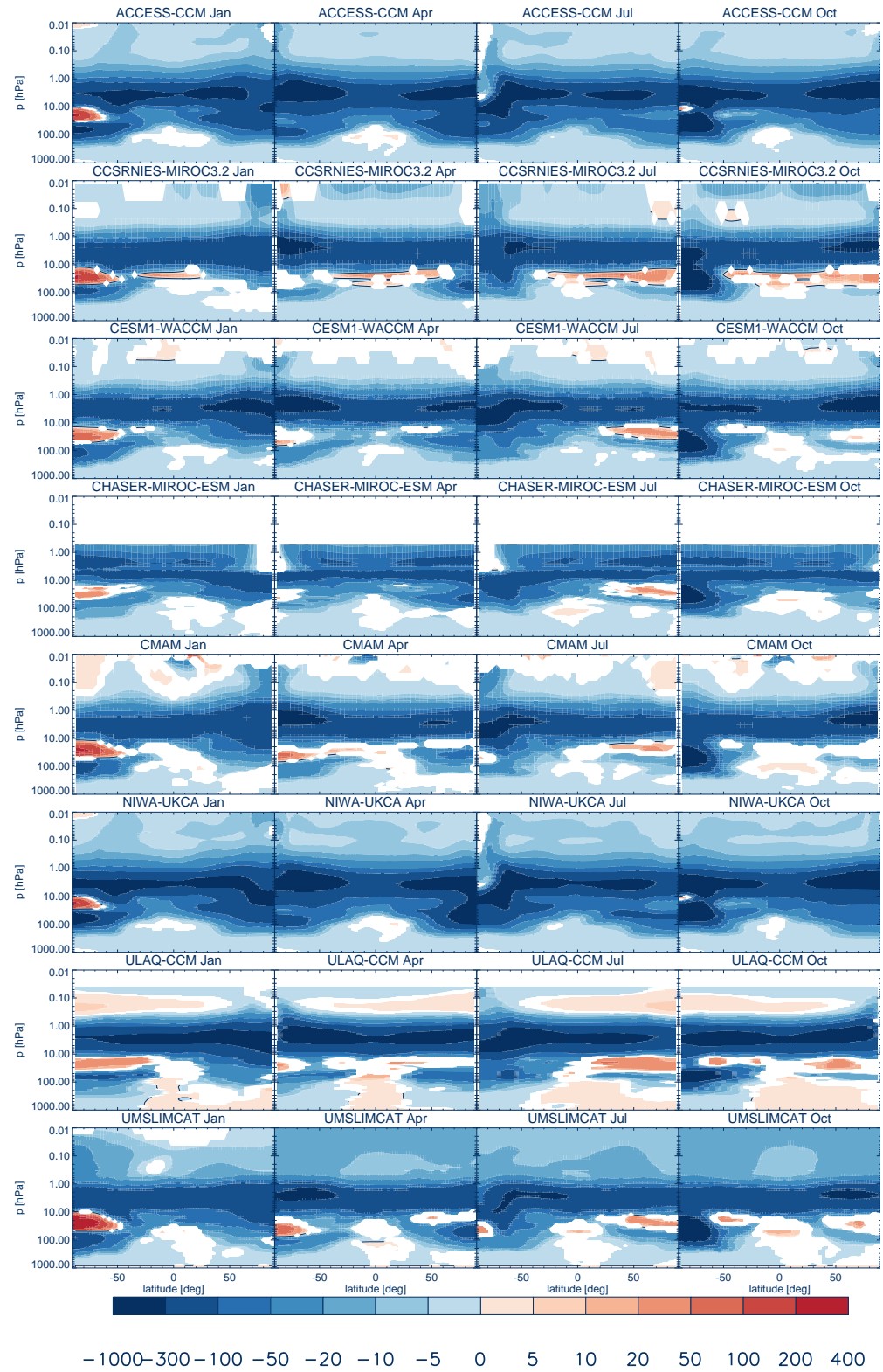

Figure 7: Same as figure 1 but for Cl$^{eq}$.

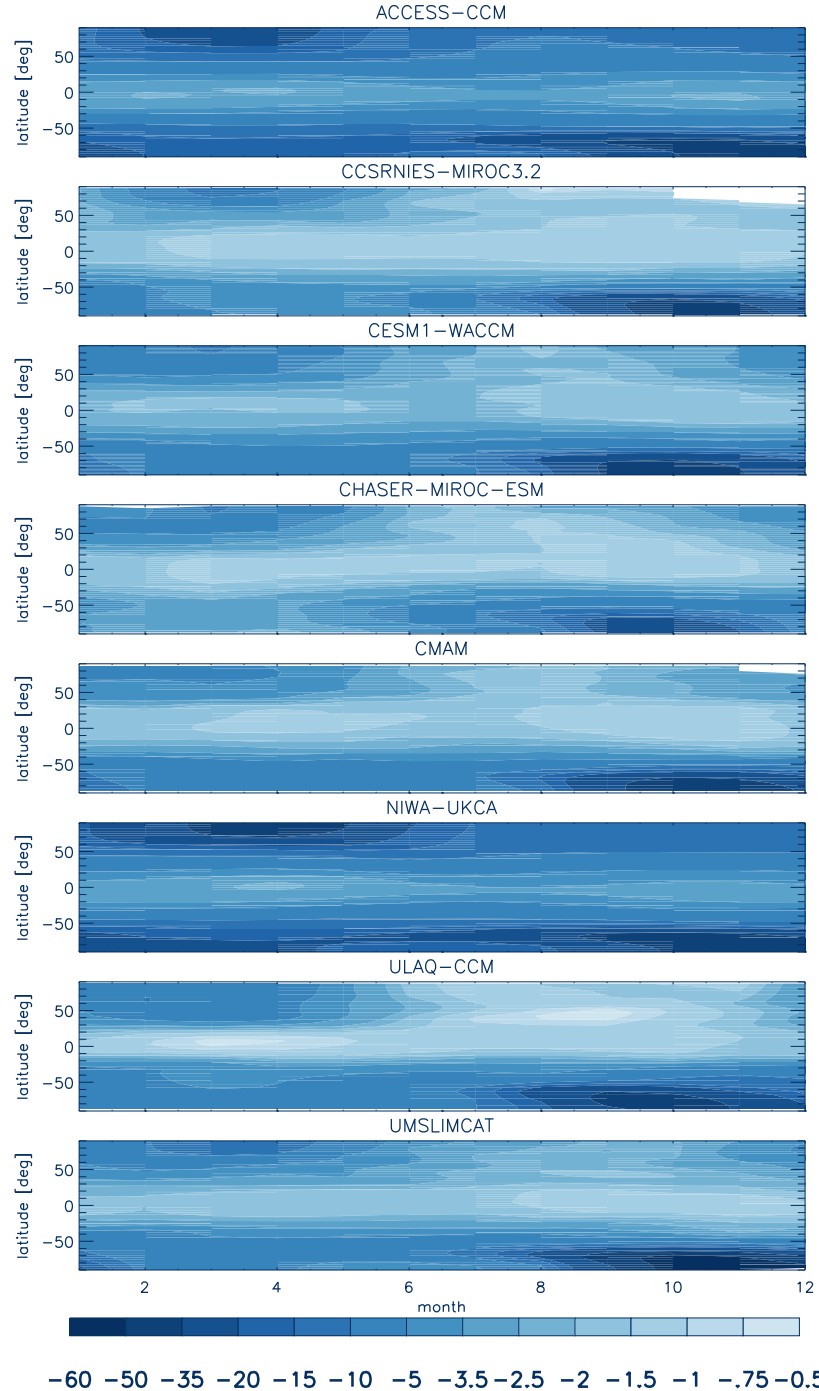

Figure 8: Same as figure 2 but for $Cl^{eq}$, in units of DU/ppbv ($Cl^{eq}$), derived from the REF-C2 and SEN-C2-fODS simulations.

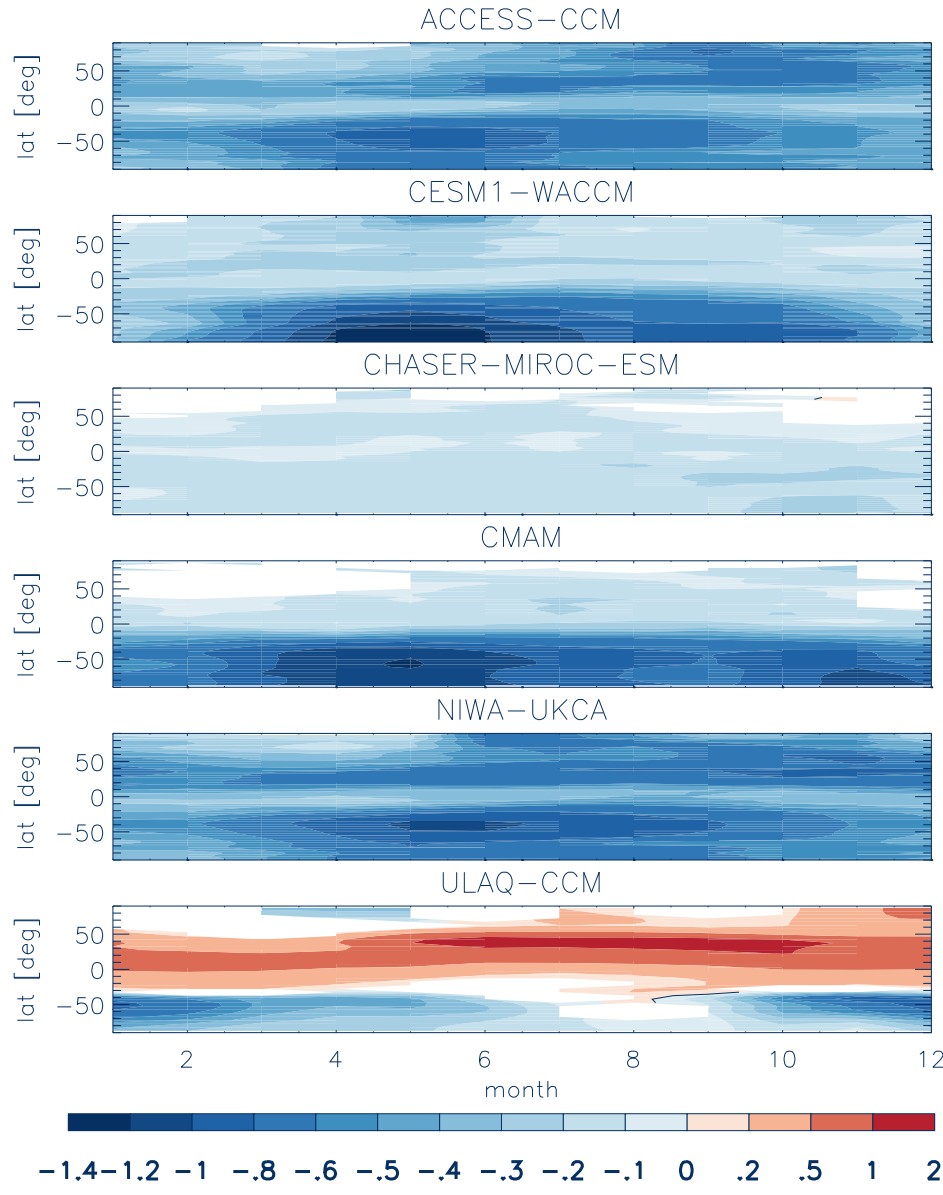

Figure 9: Ratio of zonal-mean surface ozone changes to to changes in surface $Cl^{eq}$ (in ppbv / ppbv) as derived from the REF-C2 and SEN-C2-fODS simulations.

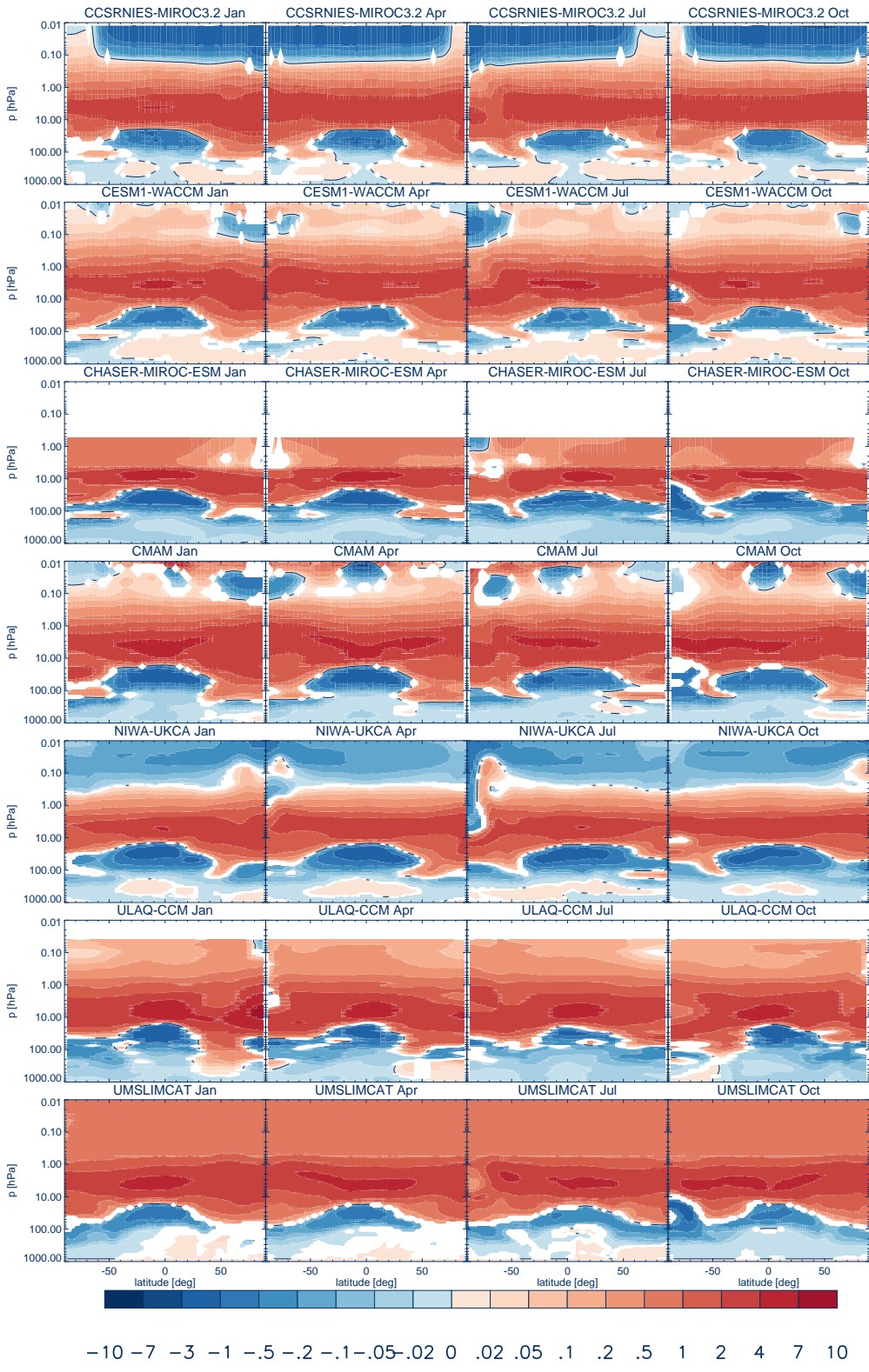

Figure 10: Same as figure 1 but for $CO_2^{eq}$. Here units are $10^{-3}$ ppmv/ppmv.

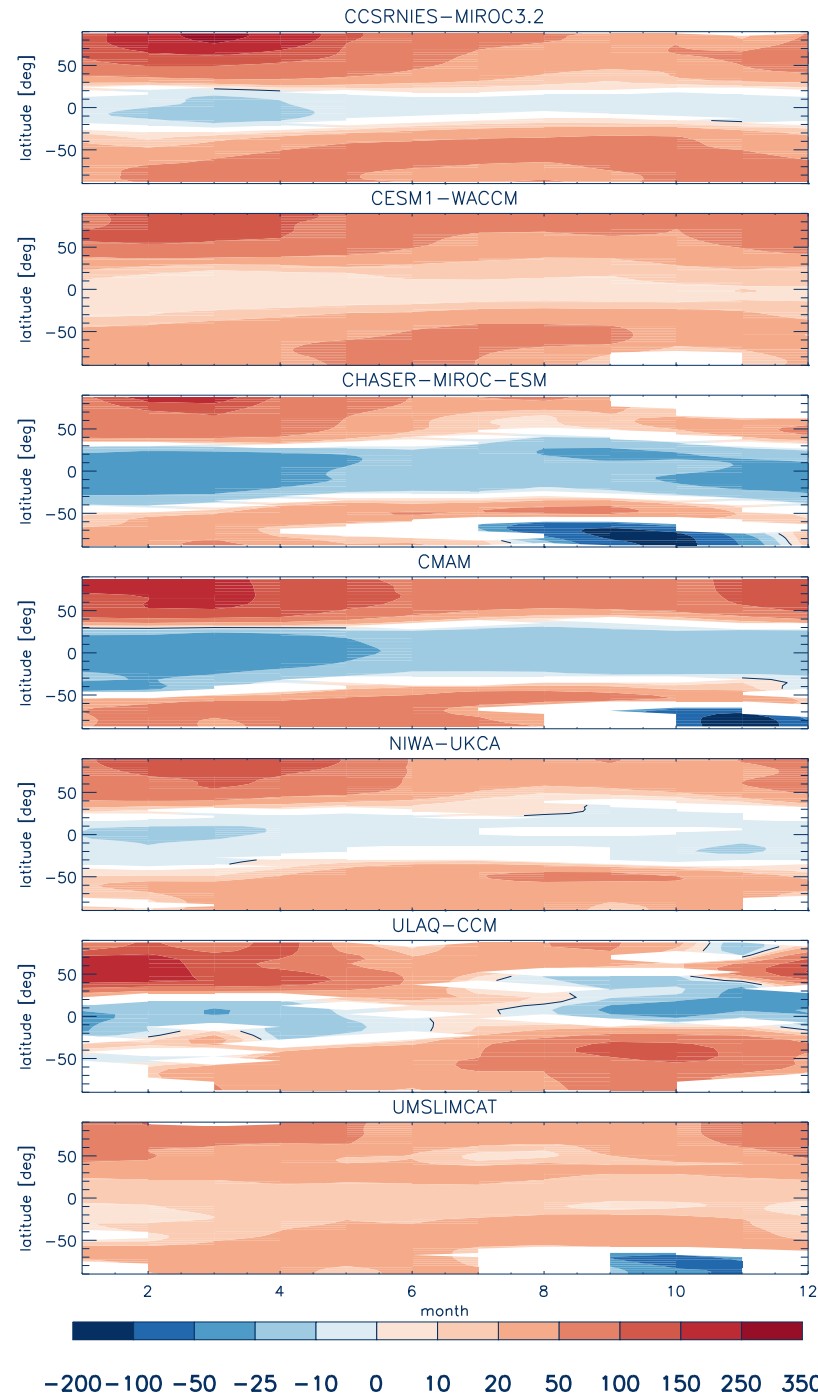

Figure 11: Same as figure 2 but for $CO_2^{eq}$, in units of $10^{-3}$ DU/ppmv ($CO_2^{eq}$), derived from the REF-C2, SEN-C2-fGHG, SEN-C2-fCH4, and SEN-C2-fN2O simulations.

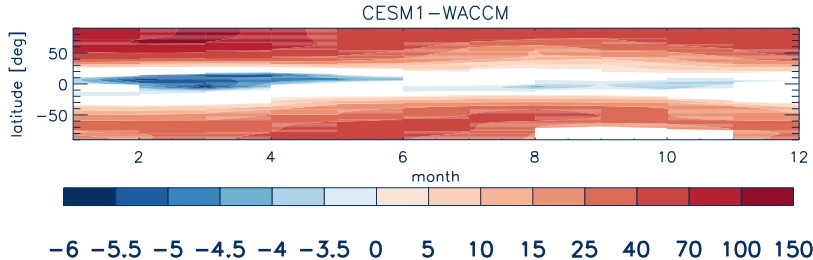

Figure 12: Same as figure 11 but for actual $CO_2$, in units of $10^{-3}$ DU/ppmv ($CO_2$), derived from the REF-C2 and fixed-$CO_2$ simulations of CESM1-WACCM.

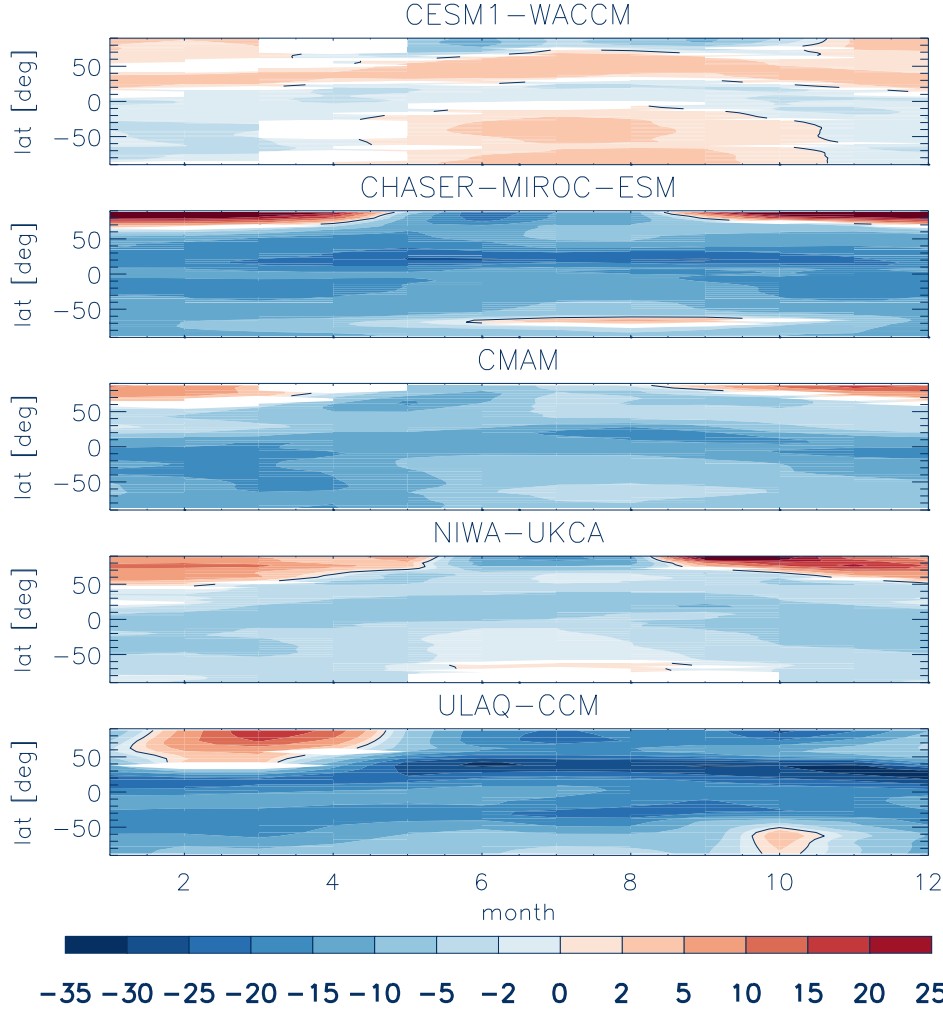

Figure 13: Ratio of zonal-mean surface ozone changes to changes in surface $CO_2^{eq}$, times $10^6$, as derived from the REF-C2, SEN-C2-fGHG, SEN-C2-fCH4, and SEN-C2-fN2O.

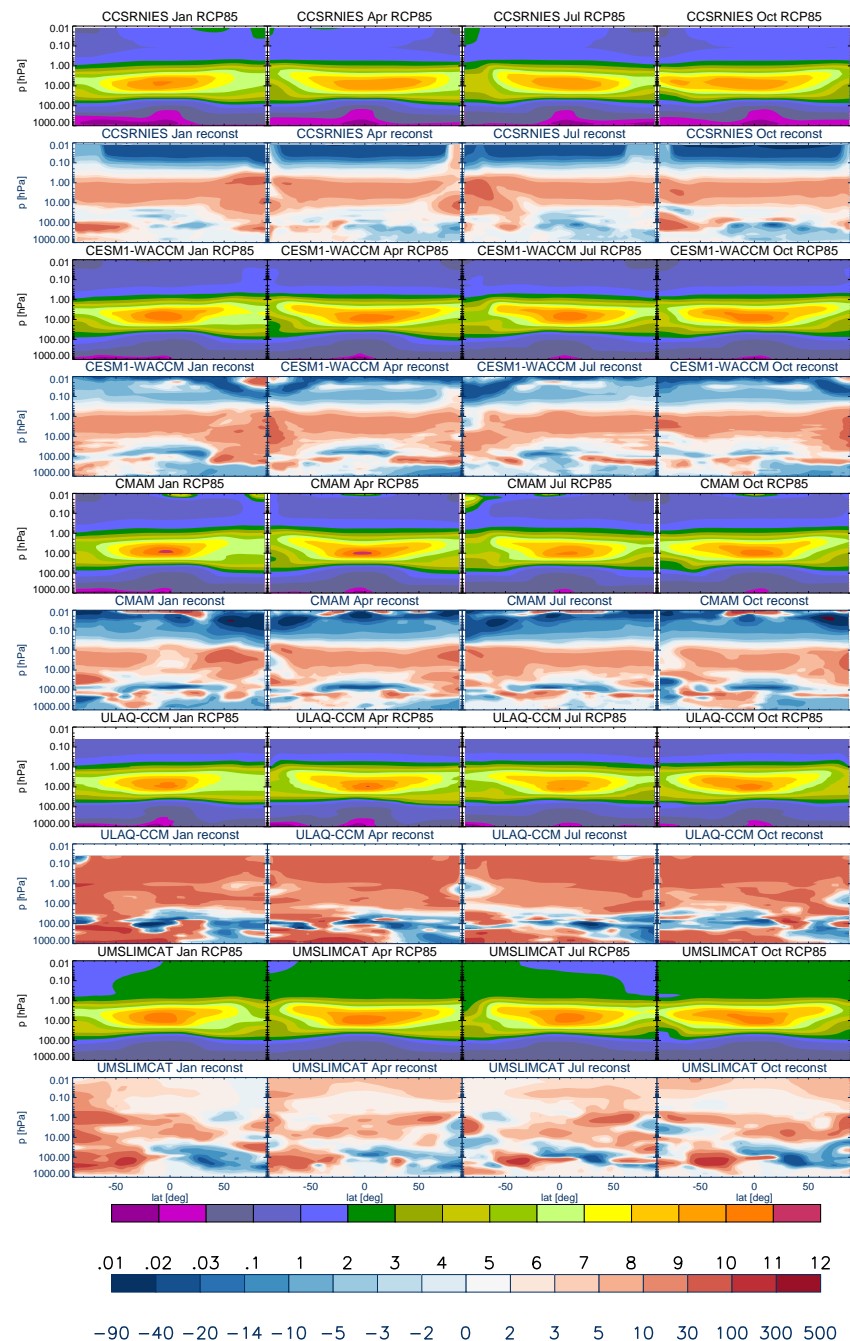

Figure 14: Rows 1, 3, 5, 7, and 9: Zonal-mean ozone (ppmv), averaged individually for the months of January, April, July, and October, for the years 2090-2099 of the RCP 8.5 scenario, as simulated by the CCSRNIES-MIROC3.2, CESM1-WACCM, CMAM, ULAQ-CCM, and UMSLIMCAT models. Rows 2, 4, 6, 8, and 10: Percentage difference between the rescaled and simulated ozone fields. The rescaling is based on the REF-C2 simulations and the $a$, $b$, and $d$ coefficients as derived versus the SEN-C2-fCH4, -fN2O, and -fGHG simulations. Note that the ODSs evolve identically in REF-C2 and in SEN-C2-RCP85.