# Peer review of "ozone-depleting substances in CCMI-1 simulations"

_Atmospheric Chemistry and Physics, 2017_

## Referee Comment (RC1) · Anonymous Referee #1 · 7 Aug 2017

This paper presents results from simulations, coordinated under the CCMI-1 initiative, performed from a number of chemistry climate models. These results have an interest to the climate community at large as they outline how the simulated ozone field in these different models is impacted by changes in a number of forcings, i.e., $CH_4$, $N_2O$, $Cl_{eq}$ and $CO_2^{eq}$.

The CCMI-1 initiative should provide ozone climatologies to climate models that use prescribed ozone fields in CMIP6 simulations, and this paper outlines the robust or non robust features of these climatologies. The paper is relatively clear in its presentation of the objectives, the method used, the results. I think that on the basis of these results

adding in this paper some recommendations with regards to the production of these climatologies would improve the interest of the paper.

I recommend publication of this paper in ACP. Please find below my comments, questions and remarks, first the more important ones and then the minor ones.

- line 26 and line : "there is a requirement for a robust mechanism...": as indicated in my summary of the paper, the paper would gain including indications for this robust mechanism.

- line 94: please describe how the various gases are grouped into $CO_2^{eq}$

- line 142: "and references therein": it would be useful to have here a synthesis of the main differences between these models that could have an impact on the results analysed in this paper.

- line 154: I would think that the comment here is somehow misleading. Even with prescribed or only partially interactive tropospheric composition there is a response of ozone in the stratosphere to surface methane changes as for instance is illustrated in Figure 1 for the CCSRNIES model. Therefore there should be a response of the total-column ozone. Please clarify this paragraph.

- line 161 equation 1: the text specifies line 171 that $\Delta CH_4$ is the global-mean methane mixing ratio. Shouldn't it rather be the global-mean surface methane mixing ratio? Please specify similarly what is $\Delta N_2O$, $\Delta Cl^{eq}$ as you in particular indicate that $Cl^{e}q$ is shifted by 4 years, and $\Delta CO_2^{eq}$.

- line 207: "relatively pronounced negative feedback" is not so clear in Figure 1 for WACCM. Please modify the comment.

- line 238: "whereas CESM1-WACCM, NIWA-UKCA, and SOCOL-3 produce partly insignificant decreases in most regions": if the change appears in white in the

figure, how can you conclude that it is a decrease or an increase? and according to figure 4, CMAM has larger areas with non significant results than NIWA-UKCA. Please amend the comments in the text.

- line 239: "In CMAM ...": I don't agree with this statement: from 100 to 1 hPa Figure 4 shows significant large decreases of ozone when N2O increases.

- in all figures with presentation of the NIWA-UKCA model please convert the vertical coordinate from km to hPa. What you will then present will be a better approximation then what readers would obtain doing it at glance in order to compare the NIWA-UKCA results with the results of the other models.

- line 341: "reductions of sea ice cover" please explicit here or in the presentation of the models which models do not use a prescribed sea ice albedo.

My minor or technical comments are the following:

- line 25: "first phase of CCMI": add "(CCMI-1)"

- line 57: "lower and middle atmosphere": please indicate a range of pressures

- line 62: correct "to due" with "due to"

- line 97: please specify the scenario

- line 112: "final section": please spefficy the section number

- line 184: "multiple simulations"

- line 278: please explicit EESC acronym

- legend figure 7 and figure 8: replace $Cl_y$ by $Cl_{eq}$

---

## Referee Comment (RC2) · Anonymous Referee #2 · 7 Sep 2017

This paper outlines a series of CCMI simulations carried out by several chemistry-climate models. The effects of $CH_4$, $N_2O$, equivalent Cl, and equivalent $CO_2$ on $O_3$ are presented in the profile, total column, and at the surface. The paper is clearly written but the repetitive organization and lack of new insights make it a slow read. More significantly, there is little attempt to explain the underlying causes of model differences presented. It is hypothesized in several places that different stratospheric transport and dynamical responses between the models is the cause of most of the differences. However, this is not diagnosed and the reader is left wondering what to conclude from this study (see below). An evaluation of the dynamical feedbacks between the models would help immensely. Some detailed exploration of the cause of peculiar behavior

for some of the largest model outliers would also greatly help the paper. Even some speculative remarks about the causes of specific outliers would add value to the paper. I cannot recommend publication of this paper without at least some attempt to explain the differences between the models.

General Comments

As stated above the lack of an attempt to explain the discrepancies between the models diminishes this paper. Given that representatives of all the modeling groups are co-authors of the paper, they should diagnose some select outliers in the simulations. As the paper presently stands, one must conclude one (or more?) of the following possibilities:

-Our chemical/dynamical understanding is incomplete (except in the middle stratosphere)

-These models include significant differences in their treatment of the chemistry, which induce different responses on ozone from the source gases

-There are errors in some of the models

-Dynamical variability is larger than the chemical effect of the source gas changes on ozone

-Differences in dynamical feedbacks are larger than the chemical signal on ozone

A more complete discussion of this is needed in the paper. The paper only mentions the last possibility with no analysis to support it. The relatively small regions that are eliminated by being outside the 95% confidence interval suggest that pure dynamical variability is not the cause of the differences (or at least that such dynamical variability is auto correlated on the timescale of a few years or longer and thus is included in the forced response). This is a bit surprising and so I'm curious how you computed the significance regions. Including an assessment of the dynamical feedbacks between the models is needed to support the assertion that these feedbacks are the likely cause of

the model differences. The differences in the chemistry could be isolated by comparing results of simulations nudged to reanalysis output but this is likely beyond the scope of this paper unless such calculations exist in the CCMI archive.

Other Comments

1. It would be useful to include the profile plots in density units in the supplement (i.e., convert to DU/km per source gas change or another similar unit). Then one could clearly see where the column changes are coming from.

2. Include somewhere the formulas used to compute the significance values used on the figures. This should include the assumptions made in arriving at the formulas. This could be in a methods section, appendix, or supplement.

3. Section 3.4: I wonder if it would be better to use $CO_2$ directly instead of equivalent $CO_2$ since $CO_2$ dominates the radiative effects of these gases in the stratosphere (as you note in lines 94-107).

---

## Author Response (AR1)

Dear Editor,

below please find our responses to the reviewers' comments as well as the annotated manuscript. In brief, the major changes versus the ACPD version are:

- We now include data from some additional models (ACCESS-CCM, CHASER-MIROC-ESM, GEOSCCM) as well as additional simulations from CESM1-WACCM and UMSLIMCAT. Consequently, we have added the following co-authors: Kane Stone, Robyn Schofield, Kengo Sudo, Luke Oman, Michael Manyin, and Daniele Visioni. The additional simulations give a more complete impression of the CCMI ensemble as a whole but do not fundamentally affect the conclusions reached in the ACPD version of the paper.

- In response to the reviewers' comments, we have added sections 4 (an analysis of age-of-air) and 6 (some general thoughts on the generation of a merged ozone dataset), and the appendix (dealing with the statistical method).

- We have expanded the supplement with four plots showing the ozone responses to the forcings in concentration units.

- All NIWA-UKCA and ACCESS-CCM results are now displayed in pressure-based coordinates.

Reviewer 2 asked for a comprehensive discussion of the causes of differences in model behaviour, at least for the more extreme outliers. Whilst our new discussion of age-of-air goes some way towards addressing that, we consider that delving into the inner workings of the models to figure out why they behave the way they do is more than this paper can deliver. The paper may well motivate such an analysis, but the results will not be forthcoming in time for this paper. However, we find that the age-of-air analysis already gives some indications about the likely causes of the differences in model behaviour.

Best regards,

Olaf Morgenstern.

**In boldface are our replies to the reviewers' comments.**

**Response to referee 1:**

35    This paper presents results from simulations, coordinated under the CCMI-1 initiative, performed from a number of chemistry climate models. These results have an interest to the climate community at large as they outline how the simulated ozone field in these different models is impacted by changes in a number of forcings, i.e., $CH_4$, $N_2O$, $Cl^{eq}$ and $CO_2^{eq}$. The CCMI-1 initiative should provide ozone

40    climatologies to climate models that use prescribed ozone fields in CMIP6 simulations, and this paper outlines the robust or non-robust features of these climatologies. The paper is relatively clear in its presentation of the objectives, the method used, the results. I think that on the basis of these results adding in this paper some recommendations with regards to the production of these climatologies would im-

45    prove the interest of the paper.
    I recommend publication of this paper in ACP.
    **We thank the reviewer for these encouraging comments. We have now added a section on the implications of the findings for generating ozone forcing fields for climate models that to not predict ozone.**

50
    Please find below my comments, questions and remarks, first the more important ones and then the minor ones.

    – line 26 and line : "there is a requirement for a robust mechanism...": As indicated in my summary of the paper, the paper would gain including indications

55       for this robust mechanism.
       **See above.**

    – line 94: please describe how the various gases are grouped into $CO_2^{eq}$.

       **This is now described in sufficient detail. Basically, the gases that make**

60       **up the RCP scenarios are weighted with their radiative efficiencies and summed up. It is worth noting that this is a diagnostic approach only. The various models considered here actually use various subsets of the gases considered here in their radiation schemes, and variably use or do not use lumping to account for those gases not included in these schemes. How-**

65       **ever, in all cases $CO_2^{eq}$ is only marginally larger than $CO_2$.**

    – line 142: "and references therein": it would be useful to have here a synthesis of the main differences between these models that could have an impact on the results analysed in this paper.

70    **We now attempt to do this. However, this is a pretty big task so this discussion remains fairly superficial. However, we now add a discussion on the sensitivity of age-of-air to the forcings studied here, which provides more insights on the possible causes for the differences in behaviour.**

75    – line 154: I would think that the comment here is somehow misleading. Even with prescribed or only partially interactive tropospheric composition there is a response of ozone in the stratosphere to surface methane changes as for instance is illustrated in Figure 1 for the CCSRNIES model. Therefore there should be a response of the total-column ozone. Please clarify this paragraph.

80    **That is correct. With prescribed ozone in the troposphere, a significant part of the response in total-column ozone is suppressed (and all of the surface ozone response). This skews the comparison of the response with the other models that have interactive tropospheric ozone. However, in response to this comment we now show the total-column response also for**
85    **the two models in question, CCSRNIES-MIROC 3.2 and UMSLIMCAT.**

      – line 161 equation 1: The text specifies line 171 that $\Delta CH_4$ is the global-mean methane mixing ratio. Shouldn't it rather be the global-mean surface methane mixing ratio? Please specify similarly what is $\Delta N_2O$, $\Delta Cl^{eq}$ as you in partic-
90    ular indicate that $Cl^{eq}$ is shifted by 4 years, and $\Delta CO_2^{eq}$.

      **Indeed. In all cases, the forcing fields are as applied at the surface. We have now replaced "global-mean" with "global surface mean".**

      – line 207: "relatively pronounced negative feedback" is not so clear in Figure 1
95    for WACCM. Please modify the comment.

      **We have now rephrased the whole paragraph; this formulation no longer appears.**

      – line 238: "whereas CESM1-WACCM, NIWA-UKCA, and SOCOL3 produce
100   partly insignificant decreases in most regions": if the change appears in white in the figure, how can you conclude that it is a decrease or an increase? And according to figure 4, CMAM has larger areas with non-significant results than NIWA-UKCA. Please amend the comments in the text.

      **We have rephrased the paragraph in response to this comment.**

105

      – line 239: "In CMAM ...": I don't agree with this statement: from 100 to 1 hPa Figure 4 shows significant large decreases of ozone when $N_2O$ increases.

**This was meant to refer to the region above 1 hPa. This detail is now added.**

110

– In all figures with presentation of the NIWA-UKCA model please convert the vertical coordinate from km to hPa. What you will then present will be a better approximation then what readers would obtain doing it at glance in order to compare the NIWA-UKCA results with the results of the other models.

115     **In all affected plots, we have interpolated the NIWA-UKCA and ACCESS-CCM data to a 126-level pressure grid, for easier comparison.**

– line 341: "reductions of sea ice cover": Please be explicit here or in the presentation of the models which models do not use a prescribed sea ice albedo.

120     **We now include a comment on coupling. This does not have a direct effect on the sea-ice albedo because both coupled and uncoupled models would take into account the albedo of shrinking sea ice.**

My minor or technical comments are the following:

125 – line 25: "first phase of CCMI": add "(CCMI-1)"
    **Done.**

– line 57: "lower and middle atmosphere": please indicate a range of pressures.
    **Done.**

– line 62: correct "to due" with "due to".
130     **Done.**

– line 97: Please specify the scenario.
    **Done.**

– line 112: "final section": Please speficy the section number.
    **Done.**

135 – line 184: "multiple simulations"
    **Done.**

– line 278: Please explain the EESC acronym.
    **"EESC" was used in error. We have replaced this with "Cleq".**

– legend figure 7 and figure 8: Replace $Cl_y$ by $Cl^{eq}$.
140     **Done.**

**Response to reviewer 2**

This paper outlines a series of CCMI simulations carried out by several chemistry-climate models. The effects of $CH_4$, $N_2O$, equivalent Cl, and equivalent $CO_2$ on $O_3$ are presented in the profile, total column, and at the surface. The paper is clearly written but the repetitive organization and lack of new insights make it a slow read. More significantly, there is little attempt to explain the underlying causes of model differences presented. It is hypothesized in several places that different stratospheric transport and dynamical responses between the models are the cause of most of the differences. However, this is not diagnosed and the reader is left wondering what to conclude from this study (see below). An evaluation of the dynamical feedbacks between the models would help immensely. Some detailed exploration of the cause of peculiar behaviour for some of the largest model outliers would also greatly help the paper. Even some speculative remarks about the causes of specific outliers would add value to the paper.

I cannot recommend publication of this paper without at least some attempt to explain the differences between the models.
**We thank the reviewer for these thoughtful comments. The "repetitive organization" was deliberate; the idea is to apply the same methodology to the four different forcings. The purpose of the paper is partly to inform the model PIs about how their models compare to others; hence the encyclopaedic approach. Completely diagnosing where the differences in model behaviour come from is beyond the scope of the paper. We are however now presenting an analysis of how age-of-air responds to the different forcings. Age is a much easier diagnostic than ozone because it only responds to transport. For $CH_4$ and $Cl^{eq}$, there are some qualitative inconsistencies in the responses which require further in-depth investigation.**

**General Comments**

As stated above the lack of an attempt to explain the discrepancies between the models diminishes this paper. Given that representatives of all the modeling groups are coauthors of the paper, they should diagnose some select outliers in the simulations.
**In at least one case, this has happened. We now present a "fixed-$CO_2$" simulation produced by CESM1-WACCM. In this model, the original analysis had indicated that CESM1-WACCM does not exhibit decreasing tropical total-column ozone in response to increasing $CO_2^{eq}$. However, this analysis had been based on the fGHG simulations, with the effects of changes in $CH_4$ and $N_2O$ added on subsequently. The new simulation shows that CESM1-WACCM does indeed decreasing TCO in the tropics in response to increasing $CO_2$, but the**

**response is weaker than in most other models. The discrepancy between the two results points at a limitation of the linear analysis conducted here.**

185

As the paper presently stands, one must conclude one (or more?) of the following possibilities:

– Our chemical/dynamical understanding is incomplete (except in the middle stratosphere).

190 **Our impression is that it is more our understanding of dynamics, as reflected in the model formulations, that is to blame. The relatively consistent response e.g. to $N_2O$ and $Cl^{eq}$ suggests that chemistry appears to be relatively well understood and simulated consistently.**

– These models include significant differences in their treatment of the chemistry, which induce different responses on ozone from the source gases.

195

**To some extent that may be the case, but fundamentally the consistent response of ozone in the middle stratosphere, where ozone is dominated by gas-phase chemistry, does confirm that chemistry appears to be relatively consistent across the models. That would not be a surprise, given that kinetics information is well established and available, as are methods to integrate the kinetics equations.**

200

– There are errors in some of the models.

**That cannot be ruled out, based on the new analysis of age-of-air. The analysis shows qualitative differences in behaviour within the CCMI ensemble that might indicate model formulation errors.**

205

– Dynamical variability is larger than the chemical effect of the source gas changes on ozone.

**Our analysis finds a lot of statistically significant signals, so we think dynamical variability is unlikely to blame here. Also the consistent responses e.g. of the ACCESS-CCM and NIWA-UKCA models (two largely identical models producing different dynamical variations) suggest that dynamical variability does not dominate the results, at least not in these cases.**

210

– Differences in dynamical feedbacks are larger than the chemical signal on ozone.

215 A more complete discussion of this is needed in the paper. The paper only mentions the last possibility with no analysis to support it. The relatively small regions that are eliminated by being outside the 95% confidence interval suggest that pure dynamical variability is not the cause of the differences (or at least that such dynamical variability is auto-correlated on the timescale of a

220 few years or longer and thus is included in the forced response). This is a bit surprising and so I'm curious how you computed the significance regions. Including an assessment of the dynamical feedbacks between the models is

needed to support the assertion that these feedbacks are the likely cause of
the model differences. The differences in the chemistry could be isolated by
comparing results of simulations nudged to reanalysis output but this is likely
beyond the scope of this paper unless such calculations exist in the CCMI
archive.

**Qualitative and quantitative differences in dynamical feedbacks indeed
exist; there now is a new section highlighting this for the age-of-air diag-
nostic. Significance is calculated using a standard approach, see text. The
trend calculation indeed assumes that the remainder $\epsilon$ in the regression
analysis (equation 1) consists of "white noise", so autocorrelation can be
assumed 0. We have tested this assumption and have mostly found this to
be the case, with some notable exceptions which may point to limitations
in the linear regression conducted here. Unfortunately the nudged simula-
tions in CCMI-1 (which do exist for some of the models) are of limited use
here because these simulations do not explore the sensitivities to long-lived
gaseous forcings, and also because they are too short (only covering 1980-
2010) and follow different scenarios. A comparison is of course possible,
but we agree this needs to be the subject of a separate study.**

Other Comments

1. It would be useful to include the profile plots in density units in the supplement
   (i.e., convert to DU/km per source gas change or another similar unit). Then
   one could clearly see where the column changes are coming from.

   **Done. We now include four such plots in the supplement.**

2. Include somewhere the formulas used to compute the significance values used
   on the figures. This should include the assumptions made in arriving at the
   formulas. This could be in a methods section, appendix, or supplement.

   **This is now spelt out in detail in the appendix.**

3. Section 3.4: I wonder if it would be better to use $CO_2$ directly instead of
   equivalent $CO_2$ since $CO_2$ dominates the radiative effects of these gases in
   the stratosphere (as you note in lines 94-107).

   **We have tried this alternative and generally find no substantial difference.
   We maintain that $CO_2^e$ is a more useful measure to use here because the
   SEN-C2-fGHG experiment is defined in terms of keeping all non-ODS
   GHGs constant, not just $CO_2$. Therefore $CO_2^e$ reflects more accurately
   what that does to radiative forcing. This is of course still a simplification
   because various subsets of the gases making up the RCP scenarios are ac-
   tually considered in the models' radiation schemes, and also models use or
   do not use lumping to account for gases not modelled explicitly. The im-
   pact of these considerations on the results presented here is small, though.**

[revised manuscript text omitted]

---

## Author Response (AR2)

Dear Paul,

Thanks a lot for accepting our paper. I have uploaded the final paper. Versus the ACPD version, there has been an increase in the co-authorship; the most recent co-author is Daniele Visioni. I suspect that he cannot be found yet in online searches.

Have a very Merry Christmas & a Happy New Year,

Olaf.